# Study on organic matter fractions in the surface microlayer in
## the Baltic Sea by spectrophotometric and spectrofluorometric
## methods
Violetta Drozdowska[1*], Iwona Wróbel[1,2], Piotr Markuszewski[1], Przemysław Makuch[1],
Anna Raczkowska[1,2], Piotr Kowalczuk[1]
[1] Institute of Oceanology Polish Academy of Science, Sopot, 81-712, Poland
[2] Centre for Polar Studies, Leading National Research Centre, 60 Będzińska Street, 41-200
Sosnowiec, Poland
*Corresponding author: Violetta Drozdowska (drozd@iopan.pl)
A revised manuscript submitted to submitted to Ocean Science and coded OS-2017-4R1,
June 10, 2017

**Abstract.** The fluorescence and absorption measurements of the samples collected from a surface microlayer (SML) and a subsurface layer (SS), a depth of 1 m were studied during three research cruises in the Baltic Sea along with hydrophysical studies and meteorological observations. Several absorption ($E_2$:$E_3$, S, $S_R$) and fluorescence (fluorescence intensities at Coble classified peaks: A, C, M, T, the ratio (M+T)/(A+C), HIX) indices of colored and fluorescent organic matter (CDOM and FDOM) helped to describe the changes in molecular size and weight as well as in composition of organic matter. The investigation allow to assess a decrease in the contribution of two terrestrial components (A and C) with increasing salinity (~1.64% and ~1.89 % in SML and ~0.78% and ~0.71 % in SS, respectively) and an increase of in-situ produced components (M and T) with salinity (~0.52% and ~2.83% in SML and ~0.98% and ~1.87% in SS, respectively). Hence, a component T reveals the biggest relative changes along the transect from the Vistula River outlet to Gdansk Deep, both in SML and SS, however an increase was higher in SML than in SS (~18.5% and ~12.3%, respectively). The ratio $E_2$:$E_3$ points to greater changes in a molecular weight of CDOM affected by a higher rate of photobleaching in SML. HIX index reflects a more advanced stage of humification and condensation processes in SS. Finally, the results reveal a higher rate of degradation processes occurring in SML than in SS. Thus, the specific physical properties of surface active organic molecules (surfactants) may modify, in a specific way, the solar light spectrum entering the sea and a penetration depth of the solar radiation. Research on the influence of surfactants on the physical processes linked to the sea surface become an important task, especially in coastal waters and in vicinity of the river mouths.

## 1 Introduction

The sea surface is a highly dynamic interface between the sea and the atmosphere (Soloviev and Lukas, 2006; Liss and Duce, 2005). The physicochemical and biological properties of a surface microlayer (SML, a surface film), are clearly and measurably different from the underlying water due to the molecules forming SML, called surfactants. Sea surface films are created by organic matter from marine and terrestrial sources: (i) dissolved and suspended products of marine plankton contained in seawater (Engel et al., 2017), (ii) terrestrial organic matter transported from land with riverine outflow (natural and synthetic) and (iii) natural oil leakages from the sea-bottom, iv) and various anthropogenic sources that includes discharge of hydrocarbons products from undersea oil and gas production, marine traffic pollution and terrestrial discharge hydrocarbons and persistent organic pollutants

(Cuncliffe et al., 2013; Engel et al., 2017). Surface films dissipate due to loss of material at
the sea surface, including microbial degradation, chemical and photo chemical processes, as
well as due to absorption and adsorption onto particulates (Liss et al., 1997). The surface
microlayer is almost ubiquitous and cover most of the surface of the ocean, even under high
turbulence conditions (Cuncliffe et al., 2013). Surface active molecules (surfactants) present
in SML may modify number of physical processes occurring in the surface microlayer:
surfactants affect the solar radiation penetration depth (Santos et al., 2012; Carlucci et al.,
1985), exchange of momentum between atmosphere and ocean by reducing the sea surface
roughness (Nightingale et al., 2000; Frew et al., 1990 ) and gas exchange between ocean and
atmosphere, impacting generation of aerosols from the sea surface (Vaishaya et al., 2012;
Ostrowska et al., 2015; Petelski et al., 2014). Therefore, research on the influence of
surfactants on the sea surface properties become an important task, especially in coastal
waters and in a vicinity of the river mouths (Maciejewska and Pempkowiak, 2015).
Surfactants comprise a complex mixture of different organic molecules of
amphiphilic and aromatic structures (with hydrophobic and/or hydrophilic heads) rich in
carbohydrates, polysaccharides, protein-like and humus (fulvic and humic) substances
(Williams et al., 1986; Ćosović and Vojvodić, 1998; Cuncliffe et al, 2011). Some dissolved
organic compounds possess, especially fulvic and humic substances, optically active parts
of molecules that absorb the light, called chromophores, (CDOM, *chromophoric* dissolved
organic matter), and fluorophores, that absorb and emit light (FDOM – fluorescent dissolved
organic matter). Due to the complexity and compositional variability of the dissolved organic
matter mixture, the absorption and fluorescence (excitation-emission matrix) spectroscopy
were found as fast and reliable available methods for detection and identification of the
dissolved organic matter in seawater (Stedmon et at, 2003; Hudson et al., 2007; Coble, 2007;
Jørgensen et al., 2011). Absorption and fluorescence spectra of specific organic compounds
groups may allow identification of sources transformations of dissolved organic matter
(Coble, 1996; Lakowicz, 2006). Several indices describing the changes of a concentration
(Blough and Del Vecchio, 2002), a molecular weight (Peuravuori and Pihlaja, 1997), a
composition of CDOM/FDOM (Stedmon and Bro, 2008; Boehme and Wells, 2006) and a
rate of degradation processes (Milori et al., 2002; Glatzel et al., 2003; Zsolnay, 2003) can be
calculated from the CDOM absorption and FDOM fluorescence excitation and emission
matrix spectra EEMs, that could be useful to study dissolved organic matter dynamics and
composition in surface micro layer. Recent advances in applications of the absorption and

fluorescence spectroscopy in environmental studies on aquatic dissolved organic matter both in fresh and marine environments and engineered water systems have been summarized in numerous text books and review papers (e.g. Coble, 2007; Hudson et al., 2007; Ishii and Boyer, 2012; Andrade-Eiroa et al., 2013; Nelson and Siegel, 2013; Coble et al., 2014; Stedmon and Nelson, 2015). The humic substances contribute significantly both to CDOM pool in the water column as well as to surfactants concentrations especially in coastal ocean, estuaries and semi-enclosed marine basin that are impacted by terrestrial runoff and marine traffic. Therefore optical methods could be used efficiently for determination of natural and anthropogenic organic surface active substances in SML (Drozdowska et al. 2013; Drozdowska et al., 2015; Pereira et al., 2016; Frew et al.,2004; Zhang et al., 2009; McKnight et al., 1997; Guéguen et al., 2007) .

Baltic Sea is a semi-enclosed marine basin with annual riverine discharge reaching ca. 0.5 $10^3$x km$^3$ of fresh water (Leppäranta and Myrberg, 2009). Maximum freshwater runoff occurs in April/May. The fresh water carries both high concentrations of CDOM (Drozdowska and Kowalczuk, 1999; Kowalczuk, 1999; Kowalczuk et al., 2010; Ylostallo et al., 2016) and substantial loads anthropogenic pollutants and inorganic nutrients (Drozdowska et al., 2002; Pastuszak et al., 2012) that stimulates phytoplankton blooms, This marine basin is also impacted by significant pollution caused by the high marine traffic (Konik and Bradtke, 2016). The main goal of this study was i) to distribution of concentration of specific CDOM/FDOM components in the SML and subsurface waters (SS - 1 m depth) in the salinity gradient along a transect from the Vistula River mouth to Gdansk Deep, Gulf of Gdansk, Baltic Sea; ii) observe the compositional changes of CDOM/FDOM derived from changes of spectral indices calculated from absorption and EEM spectra; iii) describe and iii) distinguishing processes that lead to observed differences in CDOM/FDOM concentration and composition in the SML and SS along sampled transect.

**2 Measurements**

**2.1** SML sampling

Sample collection for spectroscopic characterization of the dissolved organic matter contained in the SML and SS, that could be regarded as proxy for marine surfactants were conducted during three research cruises of r/v Oceania in April and October 2015 and in September 2016). Measurement of physical parameters of sea water and samples collection were performed at nine stations along the transect 'W' - from the mouth of the Vistula River,

W1, along the Gulf of Gdansk to the Gdansk Deep in the open sea, W9, (Figure 1). Gulf of
Gdansk is under direct influence of the main Polish river system, Vistula, which drains the
majority of Poland (Uścinowicz, 2011). Meteorological observations (wind speed and wind
direction, and a surface waves high were recorded) and CTD cast with use of the SeaBird
SBE 19 probe was performed at every station. Water samples were collected at SML and
SS. The SML sampling was carried out when the sea state was 0-4 B only, and there were
no detectable oil spills. The samples were collected from the board of the vessel (r/y
Oceania), that is about 2 m above the sea surface. The sampling was maintained about 15
minutes after anchoring, to avoid the turbulences in the surface layer caused by the screw
and ship movements. We used the Garrett Net, mesh 18, to collect the samples from the sea
surface microlayer, according to the procedure described by Garrett (1965). The mesh screen
is 50 cm x 50 cm, made of metal, and the size of holes is 1 mm, while the diameter of the
wire is 0.4 mm. Thus, the thickness of a collected microlayer is about 0.5 mm. On average,
22 such samplings were required to obtain 1 $dm^3$ of microlayer water. First, the screen was
immersed. Then, once totally immersed, the screen was left under the water until the
microlayer had stabilized. Finally, it was carefully raised to the surface in a horizontal
position at a speed of ca 5–6 cm $s^{-1}$ (Carlson 1982). Water was poured from the screen into
a polyethylene bottle using a special slit in the screen frame. In the same places the SS
samples from a depth of 1 m were taken by a Niskin bottle. Collected, unfiltered water
samples were stored in amber glass bottles in the dark at 4°C until analysis in the land based
laboratory.

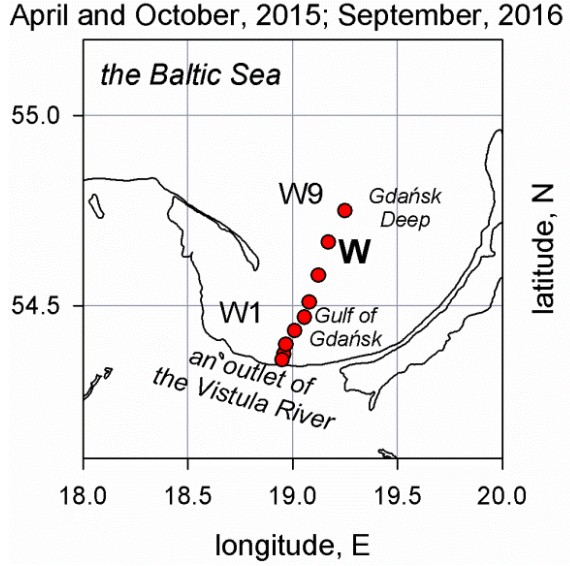


Figure 1 . Measurements stations sampled during research cruises of r/v Oceania:

28th April and 15-16th October in 2015 and 11th September in 2016.

**2.2** Laboratory spectroscopic measurements of CDOM and FDOM optical properties

Spectrophotometric and spectrofluorometric measurements of collected samples

were conducted in laboratory the Institute of Oceanology Polish Academy of Sciences,
Sopot, Poland, within a 24 h after the cruise end. Before any spectroscopic measurements
water samples were left to warm up to room temperature.

The main task in our work was to study the luminescent properties of the molecules

that form a surface microfilm. However, the seasurface microlayer is a gelatinous film
created by polysaccharides, lipids, proteins, and chromophoric dissolved organic matter
(Sabbaghzadeh et al., 2017; Cunliffe et al., 2013) and consisted of dissolved, colloidal and
particulate matter. Thus, not to dispose the absorbing and fluorescent matter involved into a
gel structure we do not filtrate the samples. In the manuscript the results of absorption and
fluorescence indices based on CDOM absorption spectra and FDOM 3D fluorescence
spectra, collected during three cruises and carried out on the unfiltered samples are
presented. There were performed the tests on filtrated and unfiltered probes, sampled during
one cruise (not published). Changes in the absorption spectra resulting from the unfiltering
of the samples occur mainly in the short UV and far VIS range. However, these differences
do not cause significant changes in the absorption indices, because they are calculated on
the basis of the shapes of the spectra (in other words: are based on the relative differences
between the values  of $a_{CDOM}(\lambda)$) in the range between the affected ends of the measuring
range. Moreover, in the studied fluorescence spectra, due to lack of filtration, we obtain a
strong elastic and non-elastic scatter band, which, however, is removed in the first step of
the analysis. The filtration procedure affects the fluorescence spectral band (Fig. 2) for a
component T (protein-like) only, that is much effectively retained on the filter, however, the
differences are the same for the SML and SS. Knowing the limitations of the applied
procedures, we decide to conduct research on unfiltered water (Ćosović and Vojvodić, 1998;
Drozdowska et al., 2015).

CDOM absorption measurements were done with use of Perkin Elmer Lambda 650

spectrophotometers in the spectral range 240 – 700. All spectroscopic measurements were
done with use of 10-cm quartz cell and ultrapure water MilliQ water was used as the
reference for all measurements. Raw recorded absorbance A(λ) spectra were processed and
the CDOM absorption coefficients $a_{CDOM}(\lambda)$ in [m$^{-1}$] were calculated by:

aCDOM(λ) = 2.303•A(λ)/l                      (1)

where, A(λ) is the corrected spectrophotometer absorbance reading at wavelength λ and l is
the optical path length in meters.

A nonlinear least squares fitting method using a Trust-Region algorithm

implemented in Matlab R2011b was applied (Stedmon et al., 2000, Kowalczuk et al., 2006)
to calculate CDOM absorption spectrum slope coefficient, S, in the spectral range 300-600
nm using the following equation:

$$a_{CDOM}(\lambda) = a_{CDOM}(\lambda_0)e^{-S(\lambda_0 - \lambda)} + K \qquad (2)$$

where: $\lambda_0$ is 350 nm, and K is a background constant that allows for any baseline shift caused
by residual scattering by fine size particle fractions, micro-air bubbles or colloidal material
present in the sample, refractive index differences between sample and the reference, or
attenuation not due to CDOM. The parameters $a_{CDOM}(350)$, S, and K were estimated
simultaneously via non-linear regression using Equation 2 in the spectral range 300-600 nm.

The organic matter fluorescence Excitation Emission matrix spectra of all collected

samples were made  using Varian Cary Eclipse scanning spectrofluorometer in a 1 cm path
length quartz cuvette using a 4 ml sample volume. A series of emission scans (280–600 nm
at 2 nm resolution) were taken over an excitation wavelength range from 250 to 500 nm at
5 nm increments. The instrument was configured to collect the signal using maximum lamp
energy and 5 nm band pass on both the excitation and emission monochromators. Prior the
measurements of each batch of samples the fluorescence EEM spectrum of Mili-Q water
blank sample was measured using the same instrumental set up. The intensity of the MiliQ
water Raman emission band was calculated by integrating the area under emission spectrum
in the spectral range: 374 - 424nm, exited at 350 nm (in literature: 355nm) (Murphy et al.,
2010). The blank MiliQ fluorescence signal was subtracted from all EEMs samples. All
blank corrected spectra were normalized to MiliQ water Raman emission (scaled to Raman
units R.U.) by dividing the resulting spectra by calculated Raman emission intensity value.

**2.3** Optical indices of CDOM and FDOM used for calculations
**2.3.1** *Absorption indices*

Based on measured absorption spectra several spectral absorption indices have been

calculated. The ratios of CDOM absorption coefficients at 250 to 365nm,
$a_{CDOM}(250)/a_{CDOM}(365)$ (called $E_2:E_3$) and at 450 to 650 nm, $a_{CDOM}(450)/a_{CDOM}(650)$,
(called $E_4:E_5$) are used to track changes in the relative size and the aromaticy of CDOM
molecules (De Haan and De Boer, 1987; Peuravuori and Pihlaja, 1997; Chin et al. 1994).
When a molecular size and aromaticy increase, the values of the ratios $E_2:E_3$ and $E_4:E_5$
decrease. This is caused by the stronger absorption at the longer wavelengths occurring due
to the presence of larger and higher molecular weighted (HMW) CDOM molecules (Helms
et al., 2008, Summers et al., 1987). In optically clear natural waters the absorption at 664 nm
is often little or immeasurable and then the absorption at 254 nm (or 280 nm) is used in lieu
of the $E_4:E_6$ ratio as an indicator of humification or aromaticy (Summers et al., 1987). The
spectral slope coefficient, *S*, of the absorption spectra, calculated in various spectral range
(Carder et al., 1989; Blough and Green, 1995) may be considered as a proxy for CDOM
composition, including the ratio of fulvic to humic acids and molecular weight (Stedmon
and Markager, 2003; Bracchini at al., 2006). The use of *S* in the narrow spectral range allows
to reveal subtle differences in the shape of the spectrum and this in turn gives insight into
the origin of organic matter (Sarpal et al., 1995). The use of narrow wavelength intervals is
advantageous as they minimize variations in *S* caused by dilution (Brown, 1977). The ratio
of the spectral slope coefficients ($S_{275\text{-}295}$ and $S_{350\text{-}400}$), $S_R$, is correlated with DOM molecular
weight (MW) and to photochemicaly induced shifts in MW (Helms et al., 2008) The spectral
slope ratio, $S_R$, was calculated as spectral slopes coefficient ratio estimated by linear fitting
of log transformed absorption spectra in the spectral ranges 275-295 nm, ($S_{275\text{-}295}$), and 350-
400, ($S_{350\text{-}400}$). Helms et al., (2008). has reported that the photochemical degradation of
terrestrial DOM lead to increase in the absolute value of the spectral slope ratio.

**2.3.2** *Fluorescence indices*

Analysis of EEM fluorescence spectra of marine waters are based on interpretation

of distinct fluorescence intensity peaks proposed by Coble (1996; Loiselle et al., 2009 ) for
different types fluorophores found in natural waters, where peak A (ex./em. 250/437 nm)
was attributed to terrestrial humic substances; peak C (ex./em. 310/429 nm) represented
terrestrial fulvic substances; peak M (ex./em. 300/387 nm) characterized marine fulvic
substances; and peak T (ex./em. 270/349 nm) represented proteinaceous substances.
Fluorescence intensities of the main FDOM components: A, C, M and T (in Raman units,
[R.U.]) were used as a proxy of FDOM concentration. A percentile contribution of the main
FDOM fluorophores, calculated as the ratio of the respective peak intensity (A, C, M or T)
to the sum (A+C+M+T) of all peak intensities, gave information about the relative changes
of a fluorophore composition in a sample (Kowalczuk et al., 2005; Drozdowska and
Józefowicz, 2015). Fluorescence intensities ratio (M+T)/(A+C) allowed to assess relative
contribution of recently in-situ produced dissolved organic matter, , (M+T) to humic
substance characterized by highly complex HMW structures (A+C) ) (Parlanti et al., 2000;
Drozdowska et al., 2015). Values of (M+T)/(A+C) ratio > 1 indicated the predominant
amount of autochthonous DOM molecules, while < 0.6 indicated the allochthonous ones.
HIX index is calculated as a ratio of fluorescence intensity at a blue part electromagnetic
radiation spectrum (435-480) (induced in) to a fluorescence intensity at the UV-C part (330–
346 nm), excited at 255 nm (Zsolnay et al., 1999). HIX index reflected the structural changes
that occurred during humification process of, causing the increase of both aromaticy (the
ratio C/H) and molecular weight of DOM molecules. Calculated spectral indices allowed to
assess DOM structural and compositional changes, and quantification of the allochthonous
(terrestrial, aromatic and highly weighted molecules) vs. autochthonous (marine humic-like
and protein-like and low molecular weighted ones) DOM fractions in the sampled transect.
**3 Results**
The SML and SS Sampling, during two research cruises, at April in 2015 and
September in 2016, was conducted in calm sea - the wind speed was almost equally to zero.
In October in 2015, fresh, north-western wind was recorded (3-4 B). This cruise started after
a week-long storm of northerly winds that caused increase of sea level at the southern part
of the Gulf of Gdansk and periodically stopped the Vistula River. As the consequence,
measured salinity along entire transect W was > 7, and values of CDOM absorption and
FDOM intensities were, even at the vicinity of the Vistula River mouth. .
**3.1** Absorption analysis
In the Baltic Sea  CDOM absorption decreases with increased salinity (Kowalczuk,
1999, Kowalczuk et al., 2006; Drozdowska and Kowalczuk, 1999), therefore as expected
CDOM absorption spectra measured at the nearest-shore station W1, are higher than
compared to those measured in outermost station W9 in the Gdansk Deep, as shown on
Figure 2. .

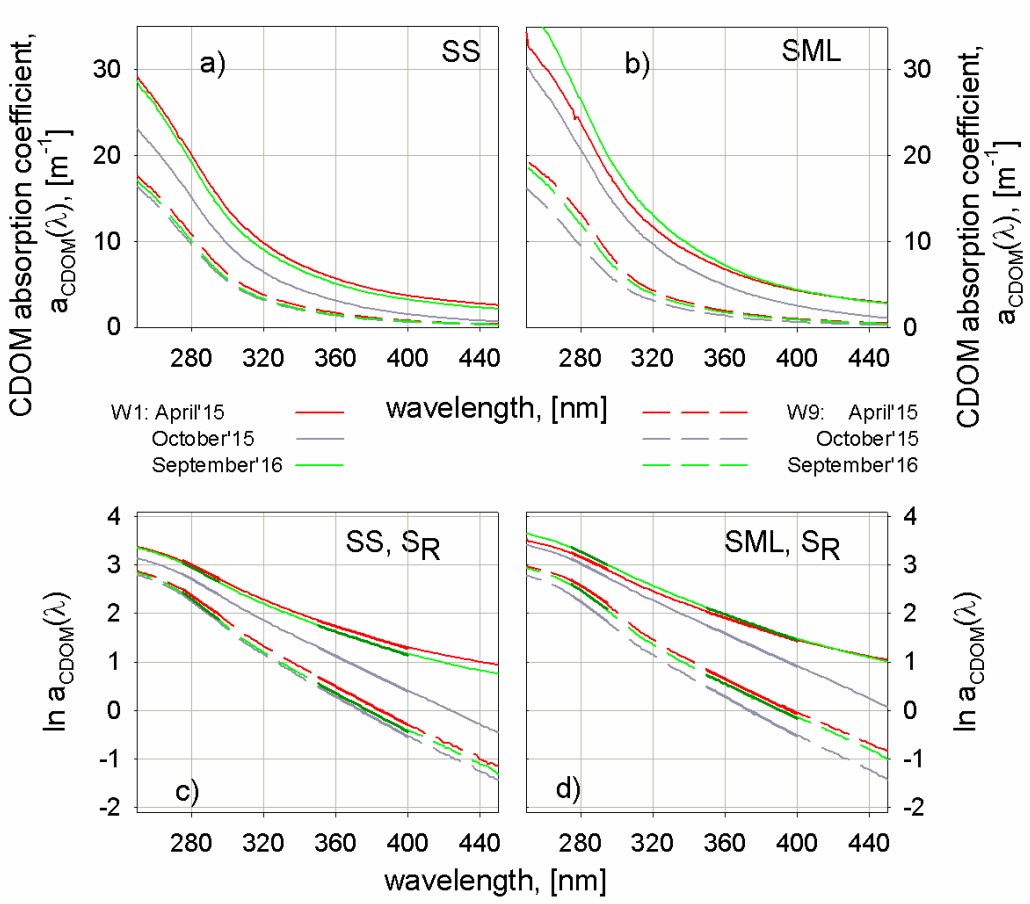


Figure 2. Absorption spectra - collected during three Baltic cruises at 28[th] April,

2015 (red lines), 15-16[th] October, 2015 (grey) and 11[th] September 2016

(green) - for W1 (solid lines) and W9 (dash lines) stations – presented in

linear scale (top panels: a, b). Natural log-transformed absorption spectra

with best-fit regression lines for two regions (275-295 nm  and 350-400

275         nm) (bottom panels: c, d).

The values of the absorption coefficient, $a_{CDOM}(\lambda)$ are the highest in the station W1,

located in the vicinity of a river outlet, and the lowest in W9, in the open sea. Moreover, the
intensity of light absorption is higher in the SML than in SS because of the enrichment effect
of the surface layer (Williams et al., 1986; Cunliffe at al., 2009), while with an increase of a
distance from the river outlet, the intensity of light absorption is decreasing significantly and
the differences between the SML and SS decrease (the calculations published in open
discussion). Furthermore, the slope ratio $S_R$, as a ratio of spectral slope coefficients in two
spectral ranges of the absorption spectra, $S_{275-295}$ and $S_{350-400}$, was calculated. The sections
of the absorption curves, marked in the appropriate narrow spectral ranges and, corresponded
to them, the values of $S_R$ are presented in Fig. 2 (c and d) and Table 1, respectively.

Table 1. Results of a slope ratio, $S_R$, for SML and SS, at W1 and W9 stations.

| A slope ratio – $S_R$ (= $S_{275-295}/S_{350-400}$) | | | | | |
|---|---|---|---|---|---|
| $S_R$ - for SS | | | $S_R$ - for SML | | |
| 28 April 2015 | 15-16 October 2015 | 11 September 2016 | 28 April 2015 | 15-16 October 2015 | 11 September 2016 |
| W1 | 1.58 | 1.16 | 1.61 | 1.43 | 1.10 | 1.35 |
| W9 | 1.30 | 1.33 | 1.40 | 1.34 | 1.35 | 1.45 |


The values of $S_R$ obtained in three cruises at W1 station (near the Vistula River outlet) were:
1.58, 1.16 and 1.61 for SS and 1.43, 1.10 and 1.35 for SML, respectively. While at W9 (open
sea) were: 1.30, 1.33 and 1.40 for SS and 1.34, 1.35 and 1.45 for SML, respectively. Hereof,
the slope ratio, $S_R$, was higher in SML than in SS in the open sea (W9), while it was opposite
in a region around the Vistula river mouth (W1). However in W9 (the open sea) the
differences were 3.1 %, 1.5 % and 3.5 %, while in W1: 10.5 %, 5.4 % and 11.9 %. The
higher values of $S_R$ in the SML in the open sea waters, mean the smaller size of CDOM that
may exist due to a photodegradation process (Helmes et al., 2008). While the lower values
of $S_R$ in the SML in the vicinity of the river outlet may mean the forming of the surface
structures from the hydrophobic molecules coming with freshwater.
Next, another absorption indices that describe the changes of molecular size/weight (the
$E_2:E_3$ ratio) and chemical composition of organic matter (a spectral slope coefficient, S),
were calculated. The results of $E_2:E3$ and S and $S_R$ in a relation to salinity are presented on
Fig. 3. The satisfying correlation between salinity and (i) the spectral slope coefficient, $S$
($r^2$=0.84 for SS and $r^2$=0.67 for SML), (ii) the slope ratio $S_R$ ($r^2$=0.58 for SS and SML) and
(iii) relative changes in the molecular weight MW ($r^2$=0.94 for SS and $r^2$=0.57 for SML)
were received. The calculations were performed by Regression Statistics, with the
Confidence interval 95 %. Moreover, the linear regression coefficients for the relations
between salinity and: S, $S_R$ and MW are, respectively 0.00439, 0.13 and 3.029 for SML and
0.00435, 0.11 and 2.293 for SS. As one can see, the linear regression coefficients achieved
higher values for SML than SS, so the processes go faster in SML than in SS.

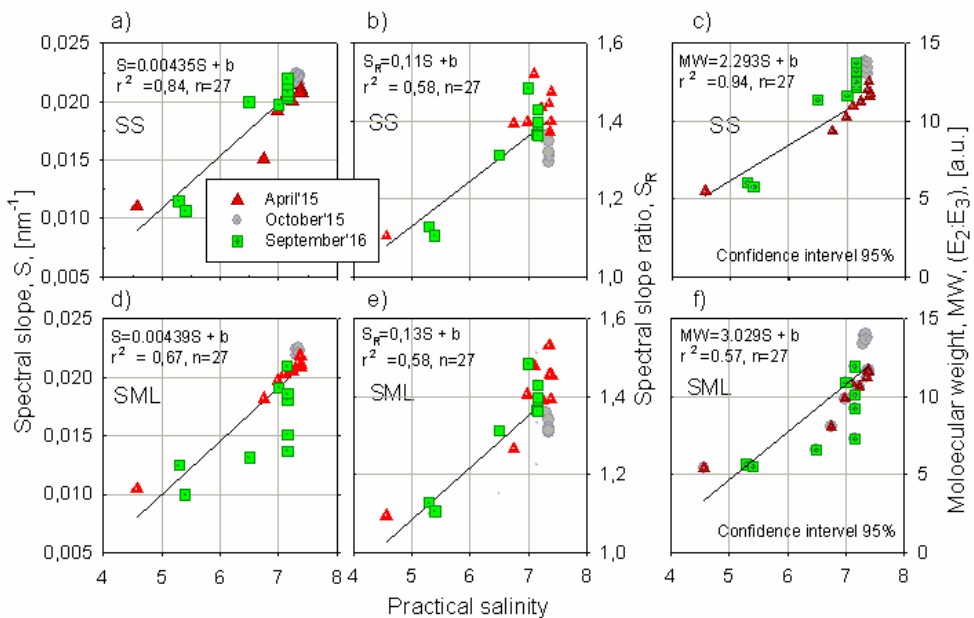


Fig. 3. The relationship between salinity and: (a) the spectral slope coefficient,
S, measured in the 300-600nm, (b) the slope ratio $S_R = S_{275-295} / S_{350-400}$,
and (c) the relative changes in the molecular weight, MW ($E_2$: $E_3$) for SS;
and: (d), (e) and (f) for SML, respectively.
Furthermore, the values of S, $S_R$ and MW are 2-, 0.5- and 3-times higher, respectively, in a
vicinity of the river outlet than in open sea.
**3.2 Fluorescence analysis**
The studies on the fluorescence properties of seawater, focused on the surface layer, were
developed in the Baltic Sea for years (Ferrari and Dowell, 1998; Drozdowska and
Kowalczuk, 2009; Drozdowska, 2007a,b) and allowed for complex analysis of the natural
components of the Baltic water (Kowalczuk et al., 2005; Stedmon et al., 2003). Based on the
analysis of 54 EEM spectra of seawater (27 samples for SML and 27 ones for SS) the
intensities of four emission bands (in [R.U.]), belonging to the main components (A, C, M
and T) of the marine CDOM were calculated. The Fig. 4 presents the 3D EEM spectra,
typical for the open sea water (the most salty) , W9, and estuarine waters (the most fresh),
W1, for the samples collected from SML and SS.

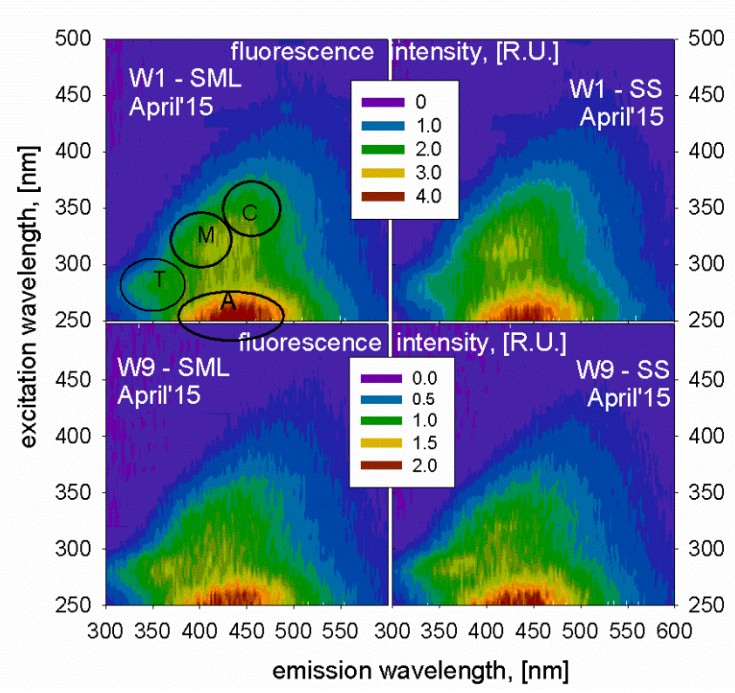

Figure 4. Examples of 3D fluorescence spectra (EEM) of the samples collected
at stations W1, near the Vistula River outlet (top panels) and W9, Gdansk
Deep (bottom panels), 28 April 2015.

The relationships between the fluorescence intensities of the main fluorescence bands (proxy
of FDOM components concentration) and salinity as well as the relative contribution of the
fluorescent components and salinity are demonstrated in Fig. 5 and 6. The changes of the
FDOM peak intensities and their relative contributions (composition of FDOM components)
in EEM were quantify by calculating the median and its percentile distribution of both the
fluorescence intensities and the relative contributions of FDOM components, for the SML
and SS in two water masses. Table 2 contains the median values of (i) fluorescence
intensities (R.U.) and (ii) percentage contribution (%) of respective peaks in the SML and
SS in two distinct water masses: one characterized by salinity <7, which is influenced by
direct fresh water discharge from Vistula River and the other characterized by salinity >7,
which is typical for open Baltic Sea waters. The ANOVA test was applied to the mentioned
median values for two cases: when the differentiation factor was (i) salinity regime and (ii)
the sampling layer. The salinity was a good factor to differentiate the variances of the median
values, while the sampling layer not. However, in spite of the p-values indicate no statistical
significance, one can see on the graphs and Table 2 that the values for the SML are always
higher than for the SS. Hence, the distinguish between the results for the SML and SS exist.
What is more, the differentiation factor is the level of sampling. The fluorescence intensities

of the main FDOM components referred to salinity demonstrate the constant linear relationships both in SS and SML (Fig. 5, upper and lower graphs, respectively). The linear regression coefficients were calculated by Regression test in Sigma Plot, with the Confident interval 95%, The linear coefficients in SML and SS, for every FDOM component, are: -1.43 and -1.02 for a component A; -0.84 and -0.65 for a component C; -0.56 and -0.43 for a component M; -0.32 and -0.3 a component T, respectively. Hence, the regression coefficients are higher in SML than in SS.

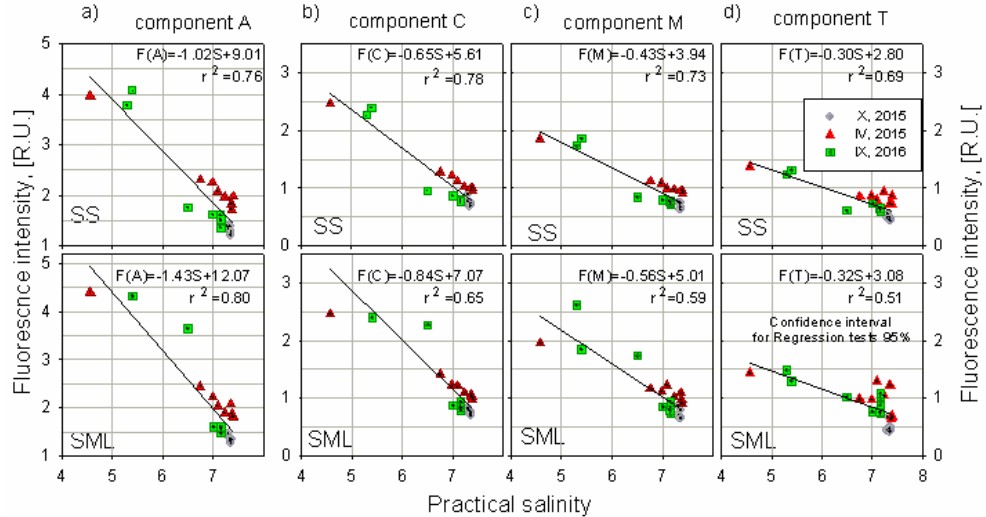

Figure. 5. Dependence of the fluorescence intensity of the main FDOM components: a) A, b) C, c) M and d) T as a linear relation to salinity for the samples from the sub-surface water (SS; top panels) and the sea surface microlayer (SML; bottom panels).

Table 2. Medians of FI[*] and PC[**] of FDOM components for coastal zone[***] and open sea waters[****]

| FDOM components | | | Salinity < 7 | | | | Salinity > 7 | | | |
|---|---|---|---|---|---|---|---|---|---|---|
| | | | A | C | M | T | A | C | M | T |
| exc./ em. (nm/nm) | | | 250/437 | 310/429 | 300/387 | 270/349 | | | | |
| fluorescence intensity, R.U. | median | SML | 2.69 | 2.27 | 1.74 | 0.98 | 1.56 | 0.84 | 0.85 | 0.69 |
| | | SS | 2.31 | 1.27 | 1.12 | 0.86 | 1.50 | 0.77 | 0.76 | 0.63 |
| percentile contribution, % | median | SML | 40.72 | 24.32 | 20.01 | 14.06 | 39.08 | 22.43 | 20.53 | 16.89 |
| | | SS | 41.52 | 22.87 | 19.92 | 14.40 | 40.75 | 22.17 | 20.90 | 16.27 |

[*]FI - a fluorescence intensity; [**]PC - a percentage contribution; [***]typical for salinity < 7; [****]typical for salinity > 7.


The percentile statistical distribution of fluorescence peak intensities in the SML and SS
layer in two water masses characterized by salinity threshold less than 7 and higher than 7,
have been presented in Fig. 6a and Fig.6b, respectively. The box-whisker plots in Fig. 6
present median values (solid line), 25th and 75th percentiles (the boundaries of the box:
closest to and farthest from zero, respectively) and 5th and 95th percentiles (whiskers below
and above the box, respectively) of the respective fluorescence intensity. There has been a
clear spatial pattern (for the coastal zone and open sea) shown on both figures that the higher
median values of A, C, M and T were observed in the SML than in SS. For salinity <7, the
median of fluorescence intensities of main FDOM components in SML were: 2.69, 2.27,
1.74 and 0.98 R.U., while in SS: 2.31, 1.27, 1.12 and 0.86 R.U. In open waters (salinity >7)
the median of fluorescence intensities of the FDOM components were in SML: 1.56, 0.84,
0.85 and 0.69 R.U., while in SS: 1.5, 0.77, 0.76 and 0.63 R.U. The median values of
respective peaks intensities are higher in SML than in SS both in coastal zone (salinity <7)
and in open sea (salinity >7). Additionally, the boundaries of the boxes show much greater
dispersion of the results in SML than in SS and greater variation in coastal zone (salinity <7)
than in open sea (salinity >7).

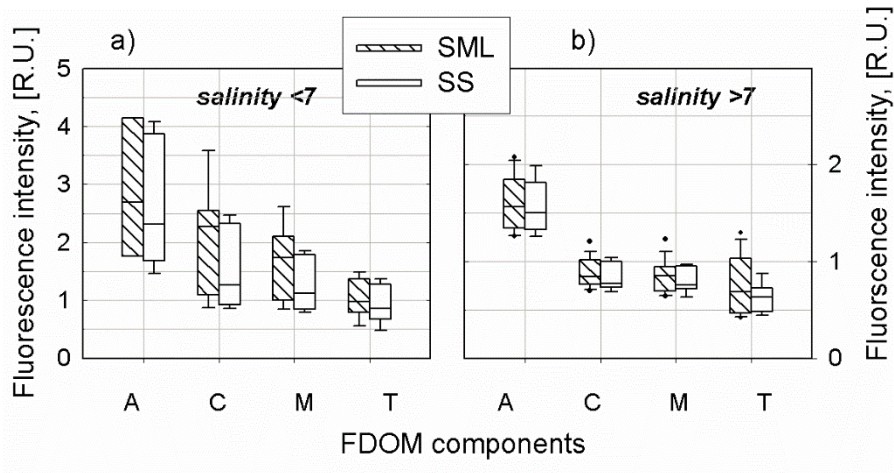


Figure 6. Dependence of the fluorescence intensity of the main FDOM
components in SML and SS as the box plots for (a) coastal water (salinity
<7) and (b) open sea (salinity >7).
The Fig. 7 shows the percentage contribution of the individual FDOM peaks calculated as
the ratio of its fluorescence intensity to the sum of the all fluorescence peak intensities (e.g.
A/(A+C+M+T)) for SS and SML samples (a left and a right graph, respectively). The box-
whisker plots in Fig. 7 present median values (solid line), 25th and 75th percentile (the
boundaries of the box: closest to and farthest from zero, respectively) and 5th and 95th
percentiles (whiskers below and above the box, respectively) of the respective percentage
contribution (a relative composition of fluorescing components of CDOM).

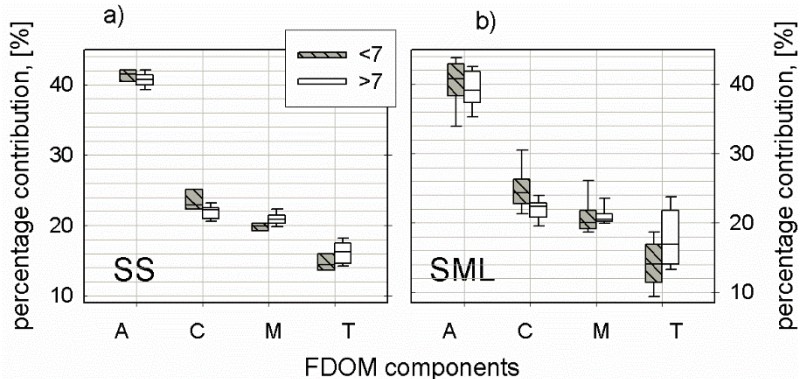


Figure. 7. Dependence of percentage contribution of the main FDOM
components as the box plots for (a) the sub-surface water, SS and (b) the
sea surface microlayer, SML; for the coastal waters (salinity <7) and
open sea (salinity >7).
For salinity <7, the medians of percentage contribution of A, C, M and T components of
marine FDOM in SML were: 40.72%, 24.32%, 20.01% and 14.06 % while in SS: 41.52%,
22.87, 19.92 and 14.40 %, respectively. In open waters (salinity >7) the median values of
FDOM components composition were in SML 39.08, 22.43, 20.53 and 16.89 % while in SS:
40.75, 22.17, 20.90 and 16.27 %. So, the contribution of two terrestrial components (A and
C) decreased with increasing salinity (~1.64% and  ~1.89 % in SML and ~0.78% and
~0.71 % in SS, respectively), while the contribution of, in-situ, in the sea produced
components (M and T) increased with salinity (~0.52% and ~2.83% in SML and ~0.98%
and  ~1.87 % in SS, respectively), Fig. 7. Considering the aforementioned changes for an
individual component in relation to its percentage contribution, the values of their relative
changes can be calculated. Hereby, the highest relative changes of the FDOM component
composition, along the transect from the Vistula River outlet to Gdansk Deep, were recorded
for component T, both in SML and SS (about 18.5 % and ~12.3 %, respectively), while the
relative changes of  A, C and M components were: 4.1, 8.1 and 2.6 % in SML and 1.9, 3.1
and 4.7 % in SS, respectively.
The values of peak intensities (A, C, M and T) allowed to calculate (i) the ratio
(M+T)/(A+C) and (ii) index HIX in SML and SS water, presented on Fig. 8.

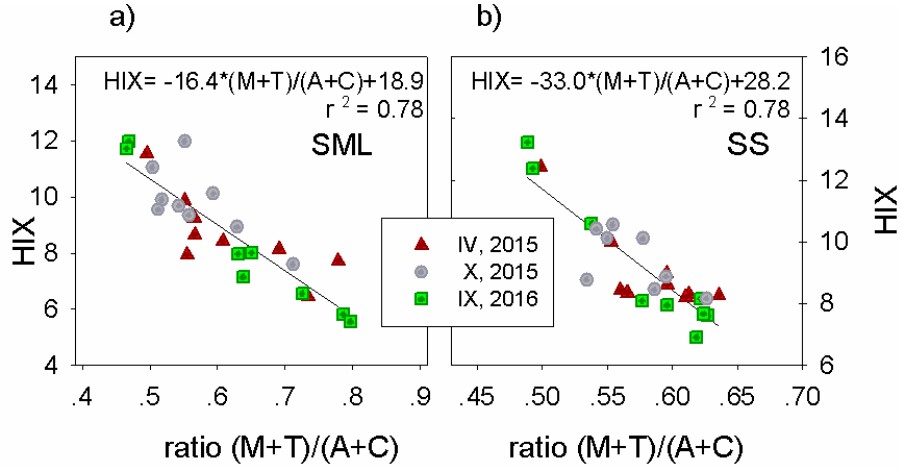

Figure 8. The relationship between the ratio (M+T)/(A+C) and HIX index for (a)

SML and (b) SS water.

The low values of the ratio (M + T)/(A + C), (<~0.6), were recorded in almost all samples from a sub-surface layer, SS, while in SML samples only from the Gulf of Gdansk. The results of the ratio varied along the transect W in the range 0.47 to 0.79 for SML and 0.49 to 0.63 for SS, from W1 to W9 respectively. Thus, the ratio describes the process that occurs more effectively in SML. The results of the index HIX achieved the higher values in the SS than in SML. What is more, the HIX index changed in SML in a range: 5.8 – 11.9 while in SS: 6.9 – 13.2. The elevated values of HIX in the SS indicate a presence of the molecules of higher molecular weight and more condensed, with higher aromaticy, in SS than in SML, Fig. 8.

**3.3** The absorption and fluorescence dependences.

The absorption and fluorescence results allow comparing the spectral slope ratio, $S_R$, with the HIX index and the ratio $E_2$:$E_3$ to find the dependences of the molecular size/weight in SML and SS with condensation degree of organic molecules and with the changes in chemical composition of organic matter, Fig. 9 (Helmes at al., 2008; Chen et al, 2011; Vähätalo and Wentzel, 2004; Zhang et al., 2013). High values of HIX index ca. 11-16, coincide with low values of $S_R$, ca. 1-1.2 (Zsolnay et al., 1999; Chari et al., 2012). While $S_{275-295} < S_{350-400}$ means the occurrence and predominance of highly condensed matter, as a dominance of and/or terrestrial DOM, with HMW molecules absorbing in a long wavelength range (Helms et al., 2008; Chen et al., 2011). Whereas, the lower HIX and higher $S_R$ values ($S_{275-295} > S_{350-400}$) mean the predominance of marine-derived, LMW molecules absorbing in a short wavelength range (Chen et al. 2011). The relation between HIX index and $S_R$ show

a simple linear relation in sub-surface waters, SS. However in the sea surface microlayer,

SML, the changes in organic matter composition, $S_R$, are not linear-related with the changes taking place in DOM molecules undergoing the degradation processes reflected by HIX values. HIX index is sensitive to the humification and condensation processes, focused on large, high weighted organic molecules, that reflect the changes in a long-wavelength range mainly (above 330 nm). However the photochemical degradation processes, resulting in a

decrease in the mass of molecules and an increase of concentration of low molecular-weighted molecules, are much more spectacular in a lower wavelength range and are held primarily in the surface microlayer, SML (Chin et al., 1994; Fuentes et al., 2006). For the same reason as was mentioned above, the relation between the ratio $E_2$:$E_3$ and $S_R$ is better correlated in SML than SS water (Helmes et al., 2008). Moreover, the relation between the

$E_2$:$E_3$ and $S_R$ (both inversely proportional to molecular size and weight) shows more discrete differences in molecular structure of the organic molecules studied in different seasons and allows to note the different nature of the water tested in October'15 (Helmes et al., 2008). The values of the ratio $E_2$:$E_3$ (inversely proportional to molecular size and weight of molecules), calculated for the data collected in October'15, point to the extremely small size

as well as almost the same size/weight of organic molecules investigated in the entirely study region both in SML and SS (De Haan and De Boer,1987; Helmes et al., 2008). That confirms a very well mixed water and the surface layer in the study area during October'15, suggested previously by the meteorological observations.

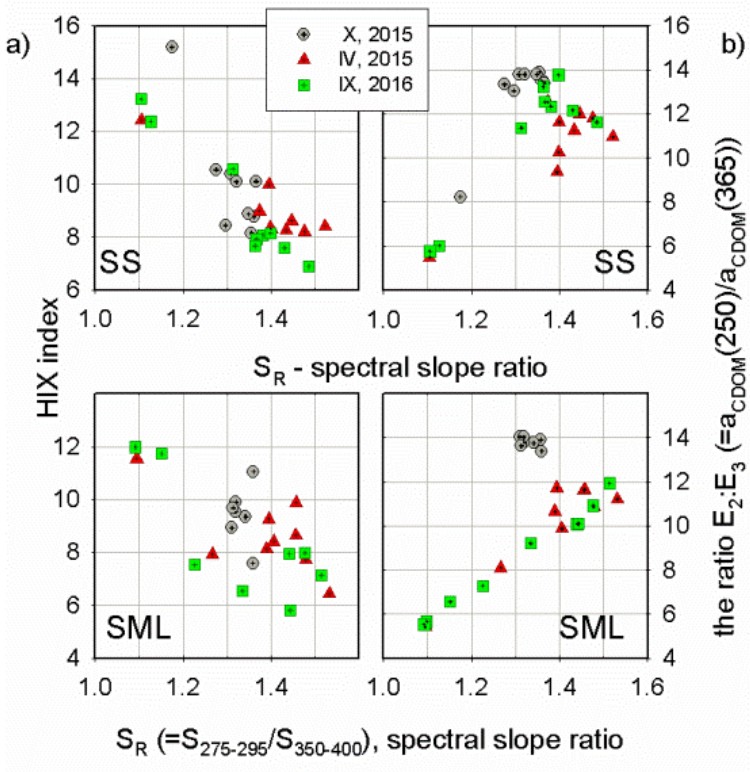

Figure 9. The relationship between the spectral slope ratio, $S_R$, and (a) HIX index
and (b) the ratio $E_2:E_3$ - for SS (top panels) and SML (bottom panels).

## 4 Discussion

The values of the absorption coefficient, $a_{CDOM}(\lambda)$ show that with an increase of a distance
from the river outlet, the intensity of light absorption (a proxy of amount of organic matter)
is decreasing significantly, both in SML and SS (Tilstone et al., 2010; Stedmon et al., 2000;
Twardowski and Donaghay, 2001; Kowalczuk et al, 1999). It shows that the main source of
CDOM in the study area is the Vistula River (Ferrari and Dowell, 1998; Kowalczuk et al.,
2005). Additionally, the higher values of the absorption were detected in the SML then in
SS, what is called the enrichment effect, that was studied for diverse range of microlayer
components in different aquatic systems (Carlson, 1982; Williams et al., 1986; Wurl et al.,
2009). Moreover, the differences between the values of the absorption coefficients calculated
for the SML and SS decrease with the increase of salinity, that was reported as the effect of
conversion POM to DOM, enhanced in the SML, by extracellular enzyme activity and export
of DOM formed in the SML to subsurface layers (Kuznetsowa and Lee; 2001; Wurl et al.,
2009) The analysis of several absorption indices (S, $S_R$ and $E_2:E_3$) reveal the changes in
composition and a decrease in molecular weight of organic matter with an increase of salinity

and a distance from the mouth of the river (Helmes at al., 2008). Molecules brought into the sea with the river waters, with increasing salinity (and time and the distance from the mouth of the river) undergo such processes as the dilution of the fresh waters in sea waters and the

480 degradation of the organic particles, induced by solar radiation (photo-bleaching) and by bacterial activity (biodegradation) (Moran et al., 2000; Helmes et al., 2008). The increase of S and $S_R$ and $E_2:E_3$ (a proxy of a decrease of molecular weight, MW) with salinity suggest a transfer of colored material from HMW fraction to the LMW fraction (Helmes et al., 2008). Moreover, the linear regression coefficients for the relations between salinity and: S, $S_R$ and

485 $E_2:E_3$ achieved higher values for SML than SS (Zhang et al., 2013). The values of the linear regression coefficients can illustrate a rate of the breakdown of large molecules to smaller ones (HMW to LMW) (Zhang et al., 2013; Timko et al., 2015; Helmes et al., 2008). They achieve the higher values in SML than in SS, thus show that in SML the dependence is stronger in the SML than in SS. Furthermore, the values of S, $S_R$ and MW, are smaller in a

490 vicinity of the river outlet about 2-, 0.5- and 3-times, respectively, than in open sea depict a presence of higher molecular weighted molecules in the estuarine waters, both in SML and SS. Hence, the higher values of $S_R$ indicate an increase of absorption in a short wavelength range (via an increase in concentration of low-weighted molecules, LWM) and a decrease of absorption in a longer wavelength range (a decrease in the concentration of big and more

condensed and high-weighted molecules, HWM) (Helmes et al., 2008; Peravuori and Pihlaja, 1997; Osburn et al., 2011). However, in a vicinity of the river mouth (W1), the studied absorption indices reached the lower values in SML than in SS. It suggests that the molecules with large molecular mass predominate in a surface microlayer. Such results may be caused by the presence of the surface slicks, visible by a naked eye, made of big surface

molecular structures. A riverine water brings into the sea a huge amount of the terrestrial amphiphilic (the molecules with hydrophobic and hydrophilic heads) organic molecules that form the surface slicks and despite the large weight of the surface molecular structures their hydrophobic properties make them float on the sea surface (Cunliffe et al., 2011).

The spectrofluorometric studies complete and confirm the absorption studies. Wherein the

505 concentration of components A, C, M and T were higher in SML than in SS in both coastal zone and open sea; the contribution of A and C components in FDOM composition decreased, while M and T increased, with an increase of salinity (Yamashita et al., 2008; McKnight et al., 2001). Moreover, the values of the fluorescence intensity of FDOM components change linearly with salinity and the linear regression coefficients show higher

values in SML than in SS (Vodacek et al., 1997; Williams et al., 2010). This may confirm a

higher rate of the degradation processes occurring in SML. The relative changes of percentage contribution of FDOM components, with an increase of salinity, depict that a component which quantity varies the most, is a fluorophore T. It may indicate on production of protein-like fluorophores caused by photobleaching and biological activity (Blough and Del Vecchio 2002;). Additionally the results of the FDOM measurements indicate that FDOM concentration is about 2-3 times higher in the coastal zone (salinity <7) than in the open sea (> 7). The results of FDOM concentration indicate the dominance of terrestrial molecules (allochthonous) in estuarine waters - due to high concentration of molecules brought by a river (A and C). The ratio (M+T)/(A+C) increased with salinity and reached the highest values in the open sea: 0.79 and 0.63 in SML and SS, respectively (Parlanti et al, 2000; Wilson and Xenopoulos, 2009; Huguet et al., 2009). Photo-degradation effect, induced by solar radiation on the molecules in a sea surface layer, results in degradation of macromolecules into particles with a lower molecular weight (i.e., a decrease of A and C and the increase the amount of molecules of lower molecular weight produced in the sea (M and T) and this process acts more rapidly in SML, (Fig. 8) (Huguet et al., 2009). The above conclusion is confirmed by the results of the ratio (M+T)/(A+C) and HIX index, which achieve respective higher and lower values in the SML than in SS due to higher fluorescence intensity at a short wavelength band belonging to marine FDOM components (M and T) (Chari et al., 2012; Stedmon and Markanger, 2003; Murphy et al., 2010; Mopper and Schults, 1993). The elevated values of HIX in the SS are an evidence of a more advance humification process of the organic molecules that make the organic molecules more condensed and with higher aromaticy (Zsolnay et al., 1999).

**5 Conclusions**

The results of the studies on the absorption and fluorescence properties of the organic matter included in the SML and SS waters are complementary. The values of the absorption coefficients as well as the fluorescence intensity give information about the decline in the CDOM/FDOM concentration with increasing salinity, both in SS and SML, however the values of the absorption and fluorescence indices indicate on the enrichment effect in in the surface microlayer. Moreover, a decreasing of DOM concentration with salinity occurs faster in SML than in SS. Analysis of absorption and fluorescence spectra allow the detection of subtle changes in the percentage composition of CDOM/FDOM components that revealed an increase of M and T (produced in-situ, in the sea) and a simultaneous decrease in A and

C (terrestrial origin) with increasing salinity. Moreover the changes of the dependence of a percentage composition and salinity occur in SML more rapidly than in SS. The results

suggest a higher rate of degradation processes in a surface microlayer (Drozdowska et al., 2015; Timko et al., 2015).

In addition, the analysis of indices obtained from the values of the intensity of the absorption and fluorescence of the samples enabled tracking sources and processes, which have been subjected to investigated molecules, in SML and SS. The authors: (i) confirm that the

550 processes of structural changes in molecules of HMW to LMW, due to effects of photo- and biodegradation, occur faster in SML than in SS (Helmes et al., 2008); (ii) organic molecules contained in a surface microlayer, SML, have a smaller molecular mass than SS, thus, SML and SS are characterized by different percentage distributions of the main FDOM components (Helmes et al., 2008; Engel et al., 2017; (iii) the fresh water of the Vistula River

is the main driving force of allochthonous character of organic matter in coastal waters of Gulf of Gdansk.

Summarizing, the distributions of light intensity reached over or behind the sea surface is modified effectively by the specific absorption and/or emission of a light by surfactants. The degradation processes of the organic molecules contained in SML and SS proceed at

560 different rates. Hence, the DOM molecules included in the SML can specifically modify the physical processes associated with the sea surface layer. It should be necessary to continue a study on the physical properties of surface microlayer in other Baltic Sea sites and in less urbanized and more natural and pristine region, like Arctic.

**Acknowledgment**

The work described in this paper was supported by a grant of ESA (European Space Agency) OCEAN FLUX, No 502-D14IN010. We also acknowledge the support by the funds of the Leading National Research Centre (KNOW) received by the Centre for Polar Studies for the period 2014-2018.

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
