# Peer review of "Study on organic matter fractions in the surface microlayer in ## the Baltic Sea by spectrophotometric and spectrofluorometric"

_Ocean Science, 2017_

## Referee Comment (RC1) · Anonymous Referee #1 · 21 Apr 2017

The authors present an interesting data set on fluorescence and absorption measurements in the sea-surface microlayer. Such measurements are valuable as they are scarce and important to understand light penetration and photochemical processes at the sea surface. Unfortunately the authors do not discuss their results to those important processes at the sea surface.

General comments: More detailed description of sampling methodology, most critical in research on the SML, are needed, and potential impacts on the results (i.e. by collected directly down the vessel's side and rather thick layers). Major data analysis

to support the conclusions are lacking from the manuscript, and statistics are partly incorrect. Discussion have to be re-written, i.e. in terms of light penetration and photochemical processes, and most importantly with references to the literature. Overall, the manuscripts requires major revision, and also grammar and language editing.

Specific comments:

Page 1 (pls see continous line numbers, thanks) Line 32: Inappropriate references; both are compendiums of different topics to the SML and upper surface processes

Line 34: How about anthropogenic sources?

Page 2 Line 1: " penetration of solar radiation and gas exchange,e.g. the generation of aerosols from the sea surface"...light penetration and gas exchange not directly related to aerosol formation. Confusing senstence.

line 9. Most of the surfactants are not fatty acids but carbohydrates and proteins with hydrophobic groups (see also William et al., 1986)

Line24-28: delete or shortened this sentence.

Line 30: SML already defined above. Be consistent with terms, e.g. sea surface microlayer and surface microlayer

Line 36: The authors mentioned here analysis of marine surfactant, but in fact they analyze FDOM/CDOM. Even though some surfactants share properties of CDOM/FDOM, these are two different groups of chemical compounds defined by their hydrophobic properties and light absorption. Please correct

Page 3 Line 5: Sampling the SML is critical due to its thickness of several ten's of micrometer 8see Cunlicffe et al. 2013). The authors use a particular thick mesh collecting a rather thick layer of 1 mm. The platform used for collecting is also not defined, and I need to assume it has been collected directly from the research vessel. Literature describes collectig SML directly from the SML but I doubt SML with full integrity can

be collected with this approach. My major concerns is that the authors ignore obvious sources of contamination (others than visible oil spills) and disturbance of the SML in the manuscript.

Line 14-17: Move to the section "Results"

Line 27: How about optical interferences of particulates in the samples during analysis?

Page 7 Line 8: "smaller and smaller" is meaningless. Provide numbers and statistics for the decrease a sthe information ais hard to extract from the figure

Line 18: W1 is near-shore, not open sea, correct?

Line 24-25: The authors assumes correlation and linear regression is the same. That is incorrect (please refer to textbook for statistics). In statistics, correlation is described as correlation coefficient r, and not as coefficient of determination (r2 commonly used in regression analysis). Also provide p values to describe the significance of the correlation. Linear regression requires an independent and dependent variable, which is not the case here.

Line 27-38: " the processes go faster in SML than in SS." i don't understand. What processes? Why faster? please explain.

Page 9 Line 14: see above regarding regression vs correlation

Line 16-19: Are the differences significantly differences? From figure 6, it seems some of the comparision of R.U. are not significantly different, but it requires statistical test and p values, which the authos should describe.

Figure 6 and 7: Slightly confusing as in Figure 6 authors grouped SML and SS in a single plot, but in Figure 7 grouped between < 7 PSU and > 7 PSU.

Page 12: Line 16/17: is this statistically different based on a significance level of 95%?

Page 13 - Discussion

[Figure]

Discussion is short (compared to the Results) and without a single reference to the literature. The authors need to clearly define section Results and Discussion, or combined both if guidelines of the journal allows it. More importantly, the authors need to discuss their results with findings from the literature, e.g. in terms of relevant processes at the sea surface such as light penetration and photochemistry.

---

## Referee Comment (RC2) · Anonymous Referee #2 · 4 May 2017

Overall

In the manuscript entitled "Study on organic matter fractions in the surface microlayer in the Baltic Sea by spectrophotometric and spectrofluorometric methods" authors pay attention to an important issue regarding to the influence of surfactants on the physical processes occurring in the sea surface layers. In the manuscript authors present an important results of absorption and fluorescence for samples collected from a surface microlayer (SML) and subsurface layer (SS) in the Batic Sea both in the open sea and near-shore. Based on absorption and fluorescence measurements authors deter-

mined several parameters to describe the changes of organic matter and discuss the processes occur in the sea surface layers.

The manuscript consists of 5 sections. In section 1 authors introduce the reader to the issue. In Section 2, authors present collection and characterization of samples, studied area and specification of measurements. Moreover, in this section authors present the detailed description of several specific absorption and fluorescence indices. Next, in section 3 authors present detailed description of obtained results taking into account the specific absorption and fluorescence indices and the relationships between them. Next, authors discuss obtained results in the Section 4 and finally the authors conclude the manuscript in Section 5.

The manuscript reports findings that are interesting for future work in ocean optics. Manuscript has scientific weight.

In my opinion the manuscript require several corrections to be suited for publication. The suggested corrections before the publication of the manuscript were mentioned below.

Detailed comments:

1. The name of the Section 2 "Method" should be specified. Now the name of this section suggests about description of used method, however authors describe in this section several issues: the used material, studied area and specification of measurements or specific absorption and fluorescence indices.

2. I suggest, that better would be if authors move the subsection 2.3 "CDOM and FDOM optical properties" and described this in separate section "Optical indices used for calculations" or "Optical indices of CDOM and FDOM used for calculations" with two subsections: absorption indices and fluorescence indices.

3. I think that better would be, if all data of calculated optical indices have been collected in one table.

4. The English language of the manuscript is good. However, several sentences are unclear or contain colloquial phrases, for example: page 7 line 8 "...become smaller an smaller.." better would be "...decreasing..." page 12 line 2 "...the biggest relative changes..." better would be "...the highest relative changes.." page 13 line 3 "...shorter wavelength..." better would be "...lower wavelength.." page 15 line 7 "What is more..." better would be "Moreover.."

I think that a little English correction can improve quality of the manuscript.

5. The data presented in tables are unnecessarily duplicated in the text, for example: - page 7 line 15-17 - duplicated data from Table 1 or - page 10 line 17-19 duplicated data from Table 2, instead the duplication, the authors should discuss this data.

6. Fig. 3 - incorrect legend.

7. Fig. 7 - no description of X-axis

8. Page 7 line 19 - W1 station describes the area near Vistula River outlet not open sea

9. Page 5 line 18 - it should be S(275-295) not S(274-295)?

---

## Referee Comment (RC3) · Anonymous Referee #3 · 5 May 2017

The manuscript titled "Study on organic matter fractions in the surface microlayer in the Baltic Sea by spectrophotometric and spectrofluorometric methods" by Drozdowska et al add to our knowledge about optical parameters of the microlayer and surface layer. In my opinion, it is a step towards remote sensing of microlayer properties, something that would be extremely helpful for studying its effect on air-sea interaction fluxes. I believe the manuscript documents well what and how has beed measured. I recommend publishing it after minor revision.

The open review process of the EGU journals has both advantages and disadvantages.

However, the fact that I see the previous two reviews makes it easier for me because I do not need to repeat what has already been told. So agreeing with most of the commands of my respectable anonymous peer-review colleagues, I will just comments on thing which I did not see in their comments.

The fluorescence intensities A, C M and T should be explained in the abstract. Something simple like "fluorescence intensities at Coble classification peaks" should be enough to give some hint to the reader what they are.

Units in the figures should be presented in [ ] braces.

Date format in Table 1 is certainly not something most English native speakers will recognize. Because of the US/UK dichotomy (09/11/2001 versus 11/09/2001), I suggest using month names explicitly (11 September 2001).

The hyphen in "October'2015" is not necessary (at least in two places). One uses it only to shorten the year (October '15).

I commend the authors for using unitless practical salinity (as all the relevant standards have it). However, the word "practical" should be added somewhere before salinity to make it obvious that the salinity was not absolute.

---

## Author Comment (AC1) · 5 Jun 2017

Thank you very much for reviewing the manuscript and your comments. I am grateful for the advice to which now I respond. I'm planning to make the language correction just after the review phase.

No, I will refer to your comments .

Reviewer #2:

[Figure]

" The manuscript reports findings that are interesting for future work in ocean optics. Manuscript has scientific weight. In my opinion the manuscript require several corrections to be suited for publication. The suggested corrections before the publication of the manuscript were mentioned below."

Detailed comments:

1. The name of the Section 2 "Method" should be specified. Now the name of this section suggests about description of used method, however authors describe in this section several issues: the used material, studied area and specification of measurements or specific absorption and fluorescence indices. I agree that the section 2 is about many aspects of the experimental measurements. Therefore the name of the Section 2 will be changed into "Measurements".

2. I suggest, that better would be if authors move the subsection 2.3 "CDOM and FDOM optical properties" and described this in separate section "Optical indices used for calculations" or "Optical indices of CDOM and FDOM used for calculations" with two subsections: absorption indices and fluorescence indices. The section 2.3 "CDOM and FDOM optical properties" in natural way tells, firstly, about the absorption spectra analysis and calculations of the absorption indices and then followed by a section dedicated to fluorescence. Hence, I'll divide the section 2.3 into 2 subsection 2.3.1 and 2.3.2, according to the Reviewer #2.

3. I think that better would be, if all data of calculated optical indices have been collected in one table. The results included in the tables I and II apply to other physical quantities as well as are calculated for different areas, therefore it'd be difficult to arrange one consistent table. Additionally, one table refers to the results contained in section 3.1, while Table II – section 3.2. Thus, in my opinion, it's better to leave the separate two tables.

4. The English language of the manuscript is good. However, several sentences are unclear or contain colloquial phrases, for example: page 7 line 8 "...become smaller

an smaller.." better would be "...decreasing..." page 12 line 2 "...the biggest relative changes..." better would be "...the highest relative changes.." page 13 line 3 "...shorter wavelength..." better would be "...lower wavelength.." page 15 line 7 "What is more..." better would be "Moreover.." I think that a little English correction can improve quality of the manuscript. Thank you, I corrected.

5. The data presented in tables are unnecessarily duplicated in the text, for example: - page 7 line 15-17 - duplicated data from Table 1 or - page 10 line 17-19 duplicated data from Table 2, instead the duplication, the authors should discuss this data. In my opinion, it is good when the information from the table and figures appear in the text. Especially since I emphasize that the higher values were received for SML.

6. Fig. 3 - incorrect legend. I corrected the legend.

7. Fig. 7 - no description of X-axis I put the description : "FDOM components"

8. Page 7 line 19 - W1 station describes the area near Vistula River outlet not open sea Thank you, I've made a mistake.

9. Page 5 line 18 - it should be S(275-295) not S(274-295)? Thank you, I've made a mistake.

I put the changes - according to your comments together with the changes suggested by the Reviewer #3.

Please also note the supplement to this comment:
http://www.ocean-sci-discuss.net/os-2017-4/os-2017-4-AC1-supplement.pdf
* * *

[revised manuscript text omitted]

---

## Author Comment (AC2) · 5 Jun 2017

Thank you very much for reviewing the manuscript and your comments.

I'm making corrections to Reviewer #1 so in 1-2 days I'll put my response to Reviewer #1 and it'll satisfy you comments as well.

Now, I'm refering to the comments of the Reviewer #3.

" So agreeing with most of the commands of my respectable anonymous peer-review

colleagues, I will just comments on thing which I did not see in their comments. "

1. The fluorescence intensities A, C M and T should be explained in the abstract. Something simple like "fluorescence intensities at Coble classification peaks" should be enough to give some hint to the reader what they are. I'll put into the abstract the information about naming of A, C, M and T as "fluorescence intensities at Coble classification peaks".

2. Units in the figures should be presented in [ ] braces. I put the all units in the all figures '[]' brackets, except Fig. 1.

3. Date format in Table 1 is certainly not something most English native speakers will recognize. Because of the US/UK dichotomy (09/11/2001 versus 11/09/2001), I suggest using month names explicitly (11 September 2001). The hyphen in "October'2015" is not necessary (at least in two places). One uses it only to shorten the year (October '15). Thank you, I wrote the name of date in Table 1 explicitly.

4. I commend the authors for using unitless practical salinity (as all the relevant standards have it). However, the word "practical" should be added somewhere before salinity to make it obvious that the salinity was not absolute. Thank you. I just put the name "Practical salinity" – as a description of the X axes in Fig.3 and Fig.5.

I put the corrections referring to Reviewr #2 and #3 together to the Corrected Manuscript.

Please also note the supplement to this comment:
http://www.ocean-sci-discuss.net/os-2017-4/os-2017-4-AC2-supplement.pdf

———————————————

[revised manuscript text omitted]

---

## Author Comment (AC3) · 15 Jun 2017

Violetta Drozdowska Institute of Oceanology Polish Academy of Sciences, u. Powstańców Warszawy 55, 81-712 Sopot, Poland

June 15, 2017

Dr Oliver Zielinski, Mrs Natascha Töpfer, Editors Ocean Science, Re: Response to reviewers comments on manuscript by Drozdowska et al., entitled "Title: Study on organic matter fractions in the surface microlayer in the Baltic Sea by spectrophotometric

and spectrofluorometric methods." submitted to Ocean Science and coded OS-2017-4

We thank the reviewers for their constructive comments. We have followed their guidance, and rewritten parts of the manuscript to place the work in better context. Especially the Discussion is now much improved thanks to the suggestions of the Reviewers. We have also gone through the text thoroughly to make any edits to the text to improve the flow and any grammatical errors we found that could be corrected. With this exercise we have also rewritten the key points and abstract to better highlight the main findings of this work.

The detailed comments to the Reviews are given below. After each Reviewers comments, our responses are written using different font face and and start with Response:

Detailed Response to review by Reviewer #1:

General comments

The authors present an interesting data set on fluorescence and absorption measurements in the sea-surface microlayer. Such measurements are valuable as they are scarce and important to understand light penetration and photochemical processes at the sea surface.

Response: We would like to thank Reviewer #1 for appreciation of our work. Unfortunately the authors do not discuss their results to those important processes at the sea surface. General comments: More detailed description of sampling methodology, most critical in research on the SML, are needed, and potential impacts on the results (i.e. by collected directly down the vessel's side and rather thick layers). Major data analysis to support the conclusions are lacking from the manuscript, and statistics are partly incorrect. Discussion have to be re-written, i.e. in terms of light penetration and photochemical processes, and most importantly with references to the literature. Overall, the manuscripts requires major revision, and also grammar and language editing.

Response: In the revised manuscript we have completely rewritten the Discussion section trying to link our results with physical and photochemical processes occurring at air-sea water interface. We have rewritten the Material and Methods section giving detailed information on sampling methodology in SML. We have also improved data analysis based on Reviewer #1 suggestions given in detailed remarks section. The whole revised manuscript was thoroughly rewritten with focus given to Material and Methods Results and Discussions sections. Prior to revised manuscript submission we have sent it to professional English editor to correct grammar and usage of English. Response to detailed comments by Reviewer 1.

Page 1 (pls see continuous line numbers, thanks)

Response: The continuous line numeration has been applied in the revised manuscript.

Line 32: Inappropriate references; both are compendiums of different topics to the SML and upper surface processes

Response: These references are an absolutely basic about a role of SML in various processes connected to the air-sea interactions. The books are mainly focused on the physics of aqueous molecular sublayers, however, they present the valid point of view on physics, chemistry and biology of the sea surface that are closely related. They contain the chapters on the exact topic of a sea surface microlayer with analysis of the physical phenomena like: viscosity, thermal effects of a cool and freshwater skin as well as diffusion and turbulence properties, connected with the top-layer of the sea. They describe in details the huge amount of dynamic processes going in the upper millimeters of the sea in various time and spatial scales – from micro-turbulence an fluxes to the planetary boundary layer. Such an introduction allows to explain, why the study on chemical composition of the organic molecules contained in a sea surface microlayer and the processes that influence the changes in their composition are the important issues to work with.

Line 34: How about anthropogenic sources?

Response: We agree with Reviewer #1 remarks. anthropogenic sources of dissolved organic matter have important contribution to pool of organic surface active compounds. Various human activities could lead to increased presence of both natural and synthetic surface active compounds found in SML. The sentence has been rewritten to a following form:

"Sea surface films are created by organic matter from sea marine and terrestrial sources: (i) dissolved and suspended products of marine plankton contained in seawater (citation), (ii) terrestrial organic matter that enter seawater transported from a land with riverine outflow (natural and synthetic), (iii) natural oil leakages from the seabottom and iv) various anthropogenic sources that includes discharge of hydrocarbons products from undersea oil and gas production, marine traffic pollution and terrestrial discharge hydrocarbons and persistent organic pollutants (citation)".

Line 1: " penetration of solar radiation and gas exchange, e.g. the generation of aerosols from the sea surface"...light penetration and gas exchange not directly related to aerosol formation. Confusing senstence.

Response: We agree with Reviewer #1 remarks. The questioned sentence has been rewritten to a following form: "Surface active molecules (surfactants) present in SML may modify the number of physical processes taking place occurring in the surface microlayer: among others the surfactants affect the solar radiation penetration depth (citation), exchange of momentum between atmosphere and ocean by reducing the sea surface roughness (citation) of penetration of solar radiation and gas exchange between ocean and atmosphere, , e.g. the impacting generation of aerosols from the sea surface (Vaishaya et al., 2012; Ostrowska et al., 2015; Petelski et al., 2014).."

line 9. Most of the surfactants are not fatty acids but carbohydrates and proteins with hydrophobic groups (see also William et al., 1986)

[Figure]

Response: We partially agree with Reviewer #1 remarks. In the paper by Ćosović and Vojvodić (Electroanalysis, 1998, 10 No.6) authors applied the aliphatic fatty acid as a model surface active substance to test the electrochemical technique for analysis of surface active substances in natural seawater. Secondly, a new sea surface microlayer model , developed by Sieburth (1983), showed that the lipids were no longer considered to be present in sufficient concentrations in SML (Cuncliffe et al, 2011, FEMS Microbiol Rev 35). However lipids, because of their strong hydrophobicity, can significantly influence surfactant activity of seawater surface. This is why I put the information about the important role of lipids in SML. The questioned sentence has been rewritten and the citation to Williams et al., 1986 added. The references list has been also updated.

" Surfactants comprise a complex mixture of different organic molecules of amphiphilic and aromatic structures (with hydrophobic and/or hydrophilic heads) rich in carbohydrates, polysaccharides, protein-like and humus (fulvic and humic) substances (Williams et al., 1986; Ćosović and Vojvodić, 1996; Cuncliffe et al, 2011) ."

Line24-28: delete or shortened this sentence.

Response: This sentence has been shortened and rewritten according to Reviewer #1 request.

"Recent advances in applications of the absorption and fluorescence spectroscopy in environmental studies on aquatic dissolved organic matter both in fresh and marine environments and engineered water systems have been summarized in numerous text books and review papers (e.g. Coble, 2007; Hudson et al., 2007; Ishii and Boyer, 2012; Andrade-Eiroa et al., 2013ab; Nelson and Siegel, 2013; Coble, 2014; Stedmon and Nelson, 2015)."

Line 30: SML already defined above. Be consistent with terms, e.g. sea surface microlayer and surface microlayer
Response: The term surface microlayer and its abbreviation SML has been used consistently in the revised version of the manuscript..

Line 36: The authors mentioned here analysis of marine surfactant, but in fact they analyze FDOM/CDOM. Even though some surfactants share properties of CDOM/FDOM, these are two different groups of chemical compounds defined by their hydrophobic properties and light absorption. Please correct

Response: We have assumed that CDOM and FDOM constitutes a significant fraction of organic marine surfactants in the Baltic Sea and could be regarded as a proxy of its concentrations. Therefore we have changes the first sentence in the Material and Methods section.

"Sample collection for spectroscopic characterization of the dissolved organic matter contained in the SML and SS, that could be regarded as proxy for organic marine surfactants was conducted during three research cruises of r/v 'Oceania' in April and October (two cruises) in 2015 and one in September 2016."

Line 5: Sampling the SML is critical due to its thickness of several ten's of micrometer (see Cunlicffe et al. 2013). The authors use a particular thick mesh collecting a rather thick layer of 1 mm. The platform used for collecting is also not defined, and I need to assume it has been collected directly from the research vessel. Literature describes collectig SML directly from the SML but I doubt SML with full integrity can be collected with this approach. My major concerns is that the authors ignore obvious sources of contamination (others than visible oil spills) and disturbance of the SML in the manuscript.

Response: The samples were collected from the board of the vessel (r/y Oceania), that is about 2 m above the sea surface. The sampling was maintained about 15 minutes after anchoring, to avoid the turbulences in the surface layer caused by the screw and

ship movements. We used the Garrett Net, mesh 18 (18 wires per inch), to collect the samples from the sea surface microlayer, according to the procedure described by Garrett [1965]. The screen is 60 cm x 60 cm, made of metal and the size of holes is 1 mm while the thickness of the wire is 0.4 mm. Thus, the thickness of a collected microlayer is about 0.5 mm and the efficiency is 60%. On average, 22 such samplings were required to obtain 1 dm3 of microlayer water. The following sampling procedure was established. First, the screen was immersed at an angle of 45âŮę . Then, once totally immersed, the screen was left under the water until the microlayer had stabilized. Finally, it was carefully raised to the surface in a horizontal position at a speed of ca 5–6 cm s−1 (Carlson 1982). Water was poured from the screen into a polyethylene bottle using a special slit in the screen frame.

Line 14-17: Move to the section "Results"

Response: These sentence has been moved to beginning of Results section.

Line 27: How about optical interferences of particulates in the samples during analysis?

Response: The main task in our work was to study the luminescent properties of the molecules that form a surface microfilm. As it is well known, the seasurface microlayer is a gelatinous film created by polysaccharides, lipids, proteins, and chromophoric dissolved organic matter (Sabbaghzadeh et al., 2017; Cunliffe et al., 2013). It means, they are consisted of dissolved, colloidal and particulate matter. Thus, not to dispose the absorbing and fluorescent matter involved into a gel structure we don't filtrate the samples. In the manuscript we present the results of absorption and fluorescence indices based on CDOM absorption spectra and FDOM 3D fluorescence spectra, collected during three cruises and carried out on the unfiltered samples.

Thus, the measured spectra of unfiltered water are distorted by the effects of scattering and refraction on the large molecules of particle matted. The absorption spectra are the mostly disrupted by scattering in short wavelength-UV, due to small protein-like particles, and in long wavelength - visible range, due to scattering on particle molecules of phytoplankton and the phytoplankton's absorption from at 430 nm to 670 nm). But the applied absorption indices (spectral slope and spectral slope ratio) are calculated in the spectral region that hardly overcome the discussed (mentioned above) spectral ranges. Moreover, the fluorescent spectra may be disordered by Rayleigh and Mie scattering on particle matter. However, in the first step of the 3D fluorescence spectra analysis the scattering effects are subtracted.

The test on differences between the filtered and unfiltered water were performed in September'16. The Figure 1. presents the results of CDOM absorption coefficient at seven wavelengths (290, 300, 310, 330, 355, 375 and 412 nm) for the samples collected in a) the SML and b) SS. The Figure 2. presents the dependence between salinity and FDOM intensity at main FDOM peaks, [R.U.] for a) the SML and b) SS. The Fig. 3 shows a dependence between aCDOM(355nm), [m-1] and FDOM intensity at peaks [R.U.] for a) unfiltered and b) filtered samples from the SML and SS. The absorption spectra show the differences in the values of the absorption coefficient, between filtered and unfiltered probes, from about 2 m-1 to 0.1 m-1 for estuary waters to the open sea, respectively, both, for the SML than SS. However, the absorption indices are calculated on the base of the shapes of the spectra (in other words: are based on the relative differences between the values of aCDOM(ïĄň)), therefore the filtration should not affect their results. Moreover, the filtration changes the fluorescence spectra (Fig. 2) for a component T (protein-like) only. However, the differences are the same for the SML and SS. Thus, in the future we plan preparing both, filtered and unfiltered samples for laboratory analysis, to compare the results of the absorption and fluorescent indices, calculated for both, filtered and unfiltered water probes. However, based on the tests, we assume that the results of absorption and fluorescent indices for unfiltered samples can be apply as the information about the properties of chromophoric organic matter contained in the gelatinous structures of surface biofilm.

Fig.1 Fig.1. Dependence between salinity and aCDOM(ïĄň), [m-1] for filtered and unfiltered samples from a) the SML and b) SS.
Fig.2 Fig. 2. Dependence between salinity and FDOM intensity at peaks, [R.U.] for filtered and unfiltered samples from a) the SML and b) SS.

Line 8: "smaller and smaller" is meaningless. Provide numbers and statistics for the decrease a sthe information is hard to extract from the figure

Response: To illustrate the decreasing differences between the values of absorption coefficients for SML and SS with increasing salinity, see Figure 3. Fig. 3 presents the dependence between salinity and the CDOM absorption coefficients, at several wavelengths: 254, 355 and 412 nm for\the SS and SML, upper and lower graphs, respectively. The values of aCDOM(ïĄň) decrease with salinity, in both: the SML and SS. However, the values of linear regression coefficients, for this dependence, are higher in the SML than in SS. Thus, for low salinity the values of aCDOM(ïĄň) in the SML are higher than in SS, while with increasing salinity the values of aCDOM(ïĄň) decrease with increasing salinity and the difference between the results of aCDOM(ïĄň) for the SML and SS decrease as well.

Fig.3 Figure 3. Dependences between salinity and CDOM absorption coefficient at several wavelengths.

Line 18: W1 is near-shore, not open sea, correct?

Response: Yes, station W1 is near the river outlet, while W9 is in open sea.

Line 24-25: The authors assumes correlation and linear regression is the same. That is incorrect (please refer to textbook for statistics). In statistics, correlation is described as correlation coefficient r, and not as coefficient of determination (r2 commonly used in regression analysis). Also provide p values to describe the significance of the correlation. Linear regression requires an independent and dependent variable, which is not the case here.

Response: I describe the relation between salinity and several absorption indices and use the regression coefficient to show the force of the dependency between the changes of one and the other variable. Therefore I should use "determination coefficient' when I describe the linear relation (and values of the regression coefficient) between the variables. Anyway, the correlation coefficient, and $R^2$, the coefficient of determination, are both measures of how well the regression model describes the data. R values near 1 indicate that the equation is a good description of the relation between the independent and dependent variables. In somehow I'd like to make prediction of the changes of absorption indices but my study are based on in-situ data and my database allow just working out the results and show the relations between the data. For the calculations of the linear regression of the variables I applied the confidence interval 95%, so p values were smaller than 0.05 – it means that probability of being wrong in concluding that there is an association between the variables. The smaller the P value, the greater the probability that there is an association.

Line 27-38: " the processes go faster in SML than in SS." i don't understand. What processes? Why faster? please explain.

Response: If a linear regression coefficient for a dependence described by the variables has a greater value in SML than in SS, it means that a proportion between the variables is higher in SML than in SS. And this situation means that the changes of a parameter along Axis Y vary faster with the values along Axis X, while these changes have nothing to do with time. Or, in the other words the relationship is stronger for SML and weaker in SS.

Page 9 Line 14: see above regarding regression vs correlation

Response: I write about the relation between the changes of both salinity and fluorescence intensity, emitted by the main component of marine FDOM (A, C, M and T).

Line 16-19: Are the differences significantly differences? From figure 6, it seems some of the comparision of R.U. are not significantly different, but it requires statistical test and p values, which the authos should describe.

Response: Firstly, the ANOVA test was applied for the results presented on Figure 6. The differentiation factor for the results presented on the figure is the level of sampling: SML or SS . The null hypothesis, H0, is that the levels of sampling are meaningless (irrelevant), while the alternative hypothesis, H1, is that the levels of sampling are significant. When we apply the Standard Deviation Statistics, proposed by Fisher, for significance level 95% (2 standard deviations) we obtain that the difference presented as the bars on figs 6 are statistical insignificant. Thus, from the ANOVA test we received the results that we cannot reject the null hypothesis, H0. Thus, the results presented on the figure 6 might be a completely random distribution. However, in spite of the p-values indicate no statistical significance, one can see on the graphs that the values for the SML are always higher than for the SS. Hence, the distinguish between the results for the SML and SS exist. What is more, the differentiation factor is the level of sampling. Then, we applied the ANOVA test for figure 7, where the differentiation factor for the results is salinity. The null hypothesis , H0, is that the different salinities are irrelevant (insignificant). The alternative hypothesis is that the salinity regimes are significant. The ANOVA test gives information that we can reject the null hypothesis. Thus, the salinity regimes for the results SML and SS are statistically significant.

Figure 6 and 7: Slightly confusing as in Figure 6 authors grouped SML and SS in a single plot, but in Figure 7 grouped between < 7 PSU and > 7 PSU.

Response: Figure 6 present the difference between fluorescence intensities for the SML and SS. While, in Figure 7 the results of percentile contribution are presented in different salinity regimes, for the SML and SS, separately. The left and right graphs, for SS and SML, respectively, show the wider range of changes of percentile contribution of all FDOM components in the SML than SS. The statistical significance was obtain for such a presentation of the results, where the differentiation factor was salinity. Moreover, the results of percentile composition of the main components of marine FDOM (included in Figure 7) can be presented in the same way as in the results in Figure 6 (in the Manuscript).

Fig.4 Figure 4. (Figure 7.) Dependence of percentage contribution of the main FDOM components in SML and SS as the box plots for (a) coastal waters (salinity <7) and (b) open sea (salinity >7).

Page 12: Line 16/17: is this statistically different based on a significance level of 95%?

Response: The calculations of the linear regression were made by Sigma Plot Toolbox with the confidence interval 95%. The calculations give the values 'a', 'b' and 'r2'.

Page 13 - Discussion is short (compared to the Results) and without a single reference to the literature. The authors need to clearly define section Results and Discussion, or combined both if guidelines of the journal allows it. More importantly, the authors need to discuss their results with findings from the literature, e.g. in terms of relevant processes at the sea surface such as light penetration and photochemistry.

Response: I put the changes into the manuscript.

Please also note the supplement to this comment:
http://www.ocean-sci-discuss.net/os-2017-4/os-2017-4-AC3-supplement.pdf

―――――――――――――――――――――

Fig.1. Dependence between salinity and $a_{CDOM}(\lambda)$, $[m^{-1}]$ for filtered and unfiltered samples from a) the SML and b) SS.

**Fig. 1.** fig-1

a)       b)

SML       SS

Fig. 2. Dependence between salinity and FDOM intensity at peaks, [R.U.] for filtered and unfiltered samples from a) the SML and b) SS.

**Fig. 2.** fig-2

[Figure]

Figure 3. Dependences between salinity and CDOM absorption coefficient at several wavelengths.

**Fig. 3.** fig-3

[revised manuscript text omitted]

tool methods (fast and reliable) to for detection and identifyication the of the dissolved
organic matter in seawater (Stedmon et at, 2003; Hudson et al., 2007; Coble, 2007; Jørgensen

Sformatowano: Nie Wyróżnienie

Z komentarzem [A3]: ???? – czy nie miało być aliphatic ?-
Jest ok. „Amfifilowe", to takie hydrofobowo-hydrofilowe.

Z komentarzem [A4]: Niedobre słowo – czy chodziło o
podstawnik – jeśli tak to trzeb znaleźć odpowiednie słowo w
słowniku. – W literaturze przedmiotu również używa się tego
określenia „head"

et al., 2011). - is the absorption and fluorescence (excitation-emission matrix) spectroscopy (Stedmon et at, 2003; Hudson et al., 2007; Coble, 2007). A unique structure of the energy levels of these organic molecules results in a specific spectral distribution of the light intensities absorbed and emitted by the molecules. Hence, the aAbsorption and fluorescence spectra of specific organic compounds groups may allow the identification of the sources transformations of dissolved organic matter (Coble, 1996; Lakowicz, 2006). Several indices describing the changes of a concentration (citationBlough and Del Vecchio, 2002), a molecular weight (Peuravuori and Pihlaja, 1997)citation), a composition of CDOM/FDOM (Stedmon and Bro, 2008; Boehme and Wells, 2006citation) and a rate of degradation processes (Milori et al., 2002; Glatzel et al., 2003; Zsolnay, 2003citation) can be calculated from The analysis of the CDOM absorption and 3D-FDOM fluorescence excitation and emission matrix fluorescence spectra EEMs, that could be useful to study dissolved organic matter dynamics and composition in surface micro layer. enabled to calculate several indices describing the changes of a concentration, a molecular weight, a composition of CDOM/FDOM and a rate of degradation processes of the organic matter occurring in the study surface layers.

[revised manuscript text omitted]

**Z komentarzem [A6]:** Recenzent chce dokładnego opisu poboru próbek w SML – to jest za mało.

[Figure]

Figure 1 . Measurements stations  sampled during research cruises of
r/v Oceania: 28[th] April and 15-16[th] October in 2015 and 11[th] September
in 2016.

**2.2.** *Laboratory spectroscopic measurements of CDOM and FDOM optical properties*
.

Spectrophotometric and spectrofluorometric measurements of  collected
samples were  conducted in laboratory the Institute of Oceanology Polish Academy
of Sciences, Sopot, Poland, within a 24 h after the cruise end. Before any spectroscopic
measurements water samples were left to warm up to room temperature.

The main task in our work was to study the luminescent properties of the molecules
that form a surface microfilm. However, The seasurface microlayer is
a gelatinous film created by polysaccharides, lipids, proteins, and chromophoric dissolved
organic matter (Sabbaghzadeh et al., 2017; Cunliffe et al., 2013) and  consisted of
dissolved, colloidal and particulate matter. Thus, not to dispose the absorbing and fluorescent matter involved into a gel structure we do no²t filtrate the samples. In the manuscript the results of absorption and fluorescence indices based on CDOM absorption spectra and FDOM 3D fluorescence spectra, collected during three cruises and carried out on the unfiltered samples are presented. There were performed the tests on filtrated and unfilted probes, sampled during one cruise (not published). Changes in the absorption spectra resulting from the unfiltering of the samples occur mainly in the short UV and far VIS range. However, these differences do not cause significant changes in the absorption indices, because they are calculated on the basis of the shapes of the spectra (in other words: are based on the relative differences between the values of $a_{CDOM}(\lambda)$) in the range between the affected ends of the measuring range  Moreover, in the studied fluorescence spectra, due to lack of filtration, we obtain a strong elastic and non-elastic scatter band, which, however, is removed in the first step of the analysis. The  filtration procedure affects  the fluorescence spectral band (Fig. 2) for a component T (protein-like) only, that is much effectively retained on the filter, however, the differences are the same for the SML and SSIt is well known that, filtration separates particulate fraction from dissolved and colloidal ones. On the other hand, during filtration the strongly surface active structures of organic molecules or macromolecules might be retained on the filter by sorption processes (Ćosović and Vojvodić, 1998)Therefore, the all studied samples are analyzed without filtrationSamples for absorption and fluorescence measurements were treated in the same manner.~~

CDOM absorption measurements were done with use of Perkin Elmer Lambda 650 spectrophotometers in the spectral range 240 – 700. All spectroscopic measurements were done with use of 10-cm quartz cell and ultrapure water MilliQ water was used as the reference for all measurements. Raw recorded absorbance $A(\lambda)$ spectra were processed and the CDOM absorption coefficients $a_{CDOM}(\lambda)$ in $[m^{-1}]$ were calculated by:

$$aCDOM(\lambda) = 2.303 \cdot A(\lambda)/l \qquad\qquad (1)$$

**Sformatowano:** Nie Wyróżnienie
**Sformatowano:** Nie Wyróżnienie
**Sformatowano:** Nie Wyróżnienie
**Sformatowano:** Nie Wyróżnienie
**Sformatowano:** Nie Wyróżnienie

**Z komentarzem [A7]:** Jak uwzgleniłaś rozpraszanie i absorpcję cząstek w pomiarach spektrofotometrem. Jak uwzględniłaś rozpraszanie w pomiarach fluorescencji?

where, $A(\lambda)$ is the corrected spectrophotometer absorbance reading at wavelength $\lambda$ and $l$ is
the optical path length in meters.

[revised manuscript text omitted]

(induced in ) to a fluorescence intensity at the UV-C  wavelength band (330–346 nm), excited at 255 nm (Chen et al., 2011; Zsolnay et al., 1999; Milori et al., 2002). HIX index reflecteds the structural changes that occuroccurred in theduring humification process of humification, causing an the increase in of both aromaticy (the ratio C/H) and molecular weight of DOM molecules. The applied indices enable to evaluate a relative contribution of the organic matter recently produced, in situ, in the sea (M and T / an intensity of a short-wavelength fluorescence band) and the molecules characterized by a highly complex structure (A and C / an intensity of a long-wavelength fluorescence band). Thus, the appliedCalculated spectral indices allowed you to assess whether DOM structural and compositional changes, and quantification of the the allochthonous (terrestrial, aromatic and highly weighted molecules) or vs. autochthonous (marine humic-like and protein-like and low molecular weighted ones) DOM fractions in the sampled transect.organic matter predominate (Chari et al., 2012).

**3 Results**

During The SML and SS Ssampling, induring two research cruises, at April in 2015 and September in 2016, was conducted in calm sea - the wind speed was almost equally to zero. However, iIn October in 2015, fresh, a northern western wind was recorded (3-4 B). InThis cruise October the cruise started after a week-long storm of northerly winds resultingthat caused increase of sea level at the southern part of the Gulf of Gdansk and periodically stopped the Vistula River. in the influx of water from the open sea and strong mixing of fresh with coastal and sea water.As the consequence, measured salinity along entire transect W was > 7, and That allows the explanation of the surprisingly values of CDOM absorption and FDOM intensities werelow concentrations (typical for a salinity above 7) of organic matter recorded along entirely transect W, even at the vicinity of the Vistula River mouth. of the Vistula River.

**3.1. *Absorption analysis**

Analysis of the absorption spectra enabled to calculate the absorption coefficients. The absorption at 254 nm exhibits the greater sensitivity to salinity changes than other wavelengths and will be applied as a proxy of CDOM concentration. In the Baltic Sea CDOM absorption decreases with increased salinity (Kowalczuk, 1999, Kowalczuk et al., 2006; Drozdowska and Kowalczuk, 1999), therefore as expected CDOM absorption spectra measured at the nearest-shore station W1, waree higher than compared to those measured in outermost station W9 in the Gdansk Deep, as shown on Figure, 2. presents the absorption spectra, for the nearest-shore, W1, and the most off-shore, W9, stations.

[Figure]

[Figure]

[Figure]

Figure 2. Absorption spectra - collected during three Baltic cruises at 28[th] April,
2015 (red lines), 15-16[th] October, 2015 (grey) and 11[th] September 2016
(green) - for W1 (solid lines) and W9 (dash lines) stations – presented in
linear scale (top panels: a, b). Natural log-transformed absorption spectra
with best-fit regression lines for two regions (275-295 nm  and 350-400
nm) (bottom panels: c, d).

The values of the absorption coefficient, $a_{CDOM}(\lambda)$ are the highest
in the station W1, located in the vicinity of a river outlet, and the lowest in W9, in the open
sea. Moreover, the intensity of light absorption is higher in the SML than in SS because of
the enrichment effect of the surface layer (Williams et al., 1986; Cunliffe at al., 2009), while
with an increase of a distance from the river outlet, the intensity of light absorption is
decreasing significantly and the differences between the SML and SS
decrease (the calculations published in open discussion). Furthermore, the slope ratio
$S_R$, as a ratio of spectral slope coefficients in two spectral ranges of the absorption spectra,
$S_{275-295}$ and $S_{350-400}$, was calculated. The sections of the absorption curves, marked in the
appropriate narrow spectral ranges and, corresponded to them, the values of $S_R$ are presented
in Fig. 2 (c and d) and Table 1, respectively.

Table 1.  Results of a slope ratio, $S_R$, for SML and SS, at W1 and W9 stations.

| | A slope ratio – $S_R$ (= $S_{275-295}$/$S_{350-400}$) | | | | | |
|---|---|---|---|---|---|---|
| | $S_R$ - for SS | | | $S_R$ - for SML | | |
| | 28[th] April 2015 | 15-16[th] October 2015 | 11[th] September 2016 | 28[th] April 2015 | 15-16[th] October 2015 | 11[th] September 2016 |
| W1 | 1.58 | 1.16 | 1.61 | 1.43 | 1.10 | 1.35 |
| W9 | 1.30 | 1.33 | 1.40 | 1.34 | 1.35 | 1.45 |

The values of $S_R$ obtained in three cruises at W1 station (near the Vistula River outlet) were:
1.58, 1.16 and 1.61 for SS and 1.43, 1.10 and 1.35 for SML, respectively. While at W9 (open
sea) were: 1.30, 1.33 and 1.40 for SS and 1.34, 1.35 and 1.45 for SML, respectively. Hereof,
the slope ratio, $S_R$, was higher in SML than in SS in the open sea (W9), while it was opposite

Sformatowano: Czcionka: 12 pkt, Nie Pogrubienie

Sformatowano: Czcionka: 12 pkt

Sformatowano: Wcięcie: Pierwszy wiersz:  1,25 cm, Odstęp Przed:  6 pkt, Po:  6 pkt

Sformatowano: Czcionka: 12 pkt

Sformatowano: Czcionka: 12 pkt

Sformatowano: Czcionka: 12 pkt

Sformatowano: Kolor czcionki: Automatyczny

Sformatowano: Kolor czcionki: Automatyczny

Sformatowano: Czcionka: 12 pkt

Sformatowano: Czcionka: 12 pkt

[revised manuscript text omitted]

---

## Author Response (AR1)

Violetta Drozdowska
Institute of Oceanology
Polish Academy of Sciences,
u. Powstańców Warszawy 55,
81-712 Sopot, Poland

June 15, 2017

Dr Oliver Zielinski,
Mrs  Natascha Töpfer,
Editors
Ocean Science,

Re: Response to reviewers comments on manuscript by Drozdowska et al., entitled "Title: Study on organic matter fractions in the surface microlayer in the Baltic Sea by spectrophotometric and spectrofluorometric methods." submitted to Ocean Science and coded OS-2017-4

We thank the reviewers for their constructive comments. We have followed their guidance, and rewritten parts of the manuscript to place the work in better context. Especially the Discussion is now much improved thanks to the suggestions of the Reviewers. We have also gone through the text thoroughly to make any edits to the text to improve the flow and any grammatical errors we found that could be corrected. With this exercise we have also rewritten the key points and abstract to better highlight the main findings of this work.

The detailed comments to the Reviews are given below. After each Reviewers comments, our responses are written using different font face and and start with **Response:**

Detailed Response to review by Reviewer #1:

General comments

The authors present an interesting data set on fluorescence and absorption measurements in the sea-surface microlayer. Such measurements are valuable as they are scarce and important to understand light penetration and photochemical processes at the sea surface.

**Response**: We would like to thank Reviewer #1 for appreciation of our work.

Unfortunately the authors do not discuss their results to those important processes at the sea surface. General comments: More detailed description of sampling methodology, most critical in research on the SML, are needed, and potential impacts on the results (i.e. by collected directly down the vessel's side and rather thick layers). Major data analysis to support the conclusions are lacking from the manuscript, and statistics are partly incorrect. Discussion have to be re-written, i.e. in terms of light penetration and photochemical processes, and most importantly with references to the literature. Overall, the manuscripts requires major revision, and also grammar and language editing.

**Response:** In the revised manuscript we have completely rewritten the Discussion section trying to link our results with physical and photochemical processes occurring at air-sea water interface. We have rewritten the Material and Methods section giving detailed information on sampling methodology in SML. We have also improved data analysis based on Reviewer #1 suggestions given in detailed remarks section. The whole revised manuscript was thoroughly rewritten with focus given to Material and Methods Results and Discussions sections. Prior to revised manuscript submission we have sent it to professional English editor to correct grammar and usage of English. **Response to detailed comments by Reviewer 1.**

Page 1 (pls see continuous line numbers, thanks)

**Response:** The continuous line numeration has been applied in the revised manuscript.

Line 32: Inappropriate references; both are compendiums of different topics to the SML and upper surface processes

**Response:**

These references are an absolutely basic about a role of SML in various processes connected to the air-sea interactions. The books are mainly focused on the physics of aqueous molecular sublayers, however, they present the valid point of view on physics, chemistry and biology of the sea surface that are closely related. They contain the chapters on the exact topic of a sea surface microlayer with analysis of the physical phenomena like: viscosity, thermal effects of a cool and freshwater skin as well as diffusion and turbulence properties, connected with the top-layer of the sea. They describe in details the huge amount of dynamic processes going in the upper millimeters of the sea in various time and spatial scales – from micro-turbulence an fluxes to the planetary boundary layer. Such an introduction allows to explain, why the study on chemical composition of the organic molecules contained in a sea surface microlayer and the processes that influence the changes in their composition are the important issues to work with.

**Response:** We agree with Reviewer #1 remarks. anthropogenic sources of dissolved organic matter have important contribution to pool of organic surface active compounds. Various human activities could lead to increased presence of both natural and synthetic surface active compounds found in SML. The sentence has been rewritten to a following form:

"Sea surface films are created by organic matter from sea marine and terrestrial sources: (i) dissolved and suspended products of marine plankton contained in seawater (Engel et al., 2017), (ii) terrestrial organic matter that enter seawater transported from a land with riverine outflow (natural and synthetic), (iii) natural oil leakages from the sea-bottom and iv) various anthropogenic sources that includes discharge of hydrocarbons products from undersea oil and gas production, marine traffic pollution and terrestrial discharge hydrocarbons and persistent organic pollutants (Cunliffe et l., 2013; Engel et al., 2017)".

Line 1: " penetration of solar radiation and gas exchange, e.g. the generation of aerosols from the sea surface"...light penetration and gas exchange not directly related to aerosol formation. Confusing senstence.

**Response:** We agree with Reviewer #1 remarks. The questioned sentence has been rewritten to a following form: "Surface active molecules (surfactants) present in SML may modify the number of physical processes taking place occurring in the surface microlayer: among others the surfactants affect the solar radiation penetration depth (Santos et al., 2012; Tisltone et al., 2010), exchange of momentum between atmosphere and ocean by reducing the sea surface roughness (Nightingale et al., 2000; Frew et al., 1990) of penetration of solar radiation and gas exchange between ocean and atmosphere, , e.g. the impacting generation of aerosols from the sea surface (Vaishaya et al., 2012; Ostrowska et al., 2015; Petelski et al., 2014).."

line 9. Most of the surfactants are not fatty acids but carbohydrates and proteins with hydrophobic groups (see also William et al., 1986)

**Response:** We partially agree with Reviewer #1 remarks. In the paper by Ćosović and Vojvodić (Electroanalysis, 1998, 10 No.6) authors applied the aliphatic fatty acid as a model surface active substance to test the electrochemical technique for analysis of surface active substances in natural seawater.  Secondly, a new sea surface microlayer model , developed  by Sieburth (1983), showed that the lipids were no longer considered to be present in sufficient concentrations in SML (Cuncliffe et al, 2011, FEMS Microbiol Rev 35). However lipids, because of their strong hydrophobicity, can significantly influence surfactant activity of seawater surface.  This is why I put the information about the important role of lipids in SML. The questioned sentence has been rewritten and the citation to Williams et al., 1986 added. The references list has been also updated.

" Surfactants comprise a complex mixture of different organic molecules of amphiphilic and aromatic structures (with hydrophobic and/or hydrophilic heads) rich in carbohydrates, polysaccharides, protein-like  and humus (fulvic and humic) substances (Williams et al., 1986; Ćosović and Vojvodić, 1996; Cuncliffe et al, 2011) ."

Line24-28: delete or shortened this sentence.

**Response:**  This sentence has been shortened and rewritten  according to Reviewer #1 request.

"Recent advances in applications of the absorption and fluorescence spectroscopy in environmental studies on aquatic dissolved organic matter both in fresh and marine environments and engineered water systems have been summarized in numerous text books and review papers (e.g. Coble, 2007; Hudson et al., 2007; Ishii and Boyer, 2012; Andrade-Eiroa et al., 2013ab; Nelson and Siegel, 2013; Coble, 2014; Stedmon and Nelson, 2015)."

Line 30: SML already defined above. Be consistent with terms, e.g. sea surface microlayer and surface microlayer

**Response:** The term surface microlayer and its abbreviation SML has been used consistently in the revised version of the manuscript..

Line 36: The authors mentioned here analysis of marine surfactant, but in fact they analyze FDOM/CDOM. Even though some surfactants share properties of CDOM/FDOM, these are two different groups of chemical compounds defined by their hydrophobic properties and light absorption. Please correct

**Response:** We have assumed that CDOM and FDOM constitutes a significant fraction of organic marine surfactants in the Baltic Sea and could be regarded as a proxy of its concentrations. Therefore we have changes the first sentence in the Material and Methods section.

"Sample collection for spectroscopic characterization of the dissolved organic matter contained in the SML and SS, that could be regarded as proxy for organic marine surfactants was conducted during three research cruises of r/v 'Oceania' in April and October (two cruises) in 2015 and one in September 2016."

Line 5: Sampling the SML is critical due to its thickness of several ten's of micrometer (see Cunlicffe et al. 2013). The authors use a particular thick mesh collecting a rather thick layer of 1 mm. The platform used for collecting is also not defined, and I need to assume it has been collected directly from the research vessel. Literature describes collectig SML directly from the SML but I doubt SML with full integrity can C2 OSD Interactive comment Printer-friendly version Discussion paper be collected with this approach. My major concerns is that the authors ignore obvious sources of contamination (others than visible oil spills) and disturbance of the SML in the manuscript.

**Response:** The samples were collected from the board of the vessel (r/y Oceania), that is about 2 m above the sea surface. The sampling was maintained about 15 minutes after anchoring, to avoid the turbulences in the surface layer caused by the screw and ship movements. We used the Garrett Net, mesh 18 (18 wires per inch), to collect the samples from the sea surface microlayer, according to the procedure described by Garrett [1965]. The screen is 60 cm x 60 cm, made of metal and the size of holes is 1 mm while the thickness of the wire is 0.4 mm. Thus, the thickness of a collected microlayer is about 0.5 mm and the efficiency is 60%. On average, 22 such samplings were required to obtain 1 dm$^3$ of microlayer water. The following sampling procedure was established. First, the screen was immersed at an angle of 45∘ . Then, once totally immersed, the screen was left under the water until the microlayer had stabilized. Finally, it was carefully raised to the surface in a horizontal position at a speed of ca 5–6 cm s$^{-1}$ (Carlson 1982). Water was poured from the screen into a polyethylene bottle using a special slit in the screen frame.

Line 14-17: Move to the section "Results"

**Response:** These sentence has been moved to beginning of Results section.

Line 27: How about optical interferences of particulates in the samples during analysis?

**Response**: The main task in our work was to study the luminescent properties of the molecules that form a surface microfilm. As it is well known, the seasurface microlayer is a gelatinous film created by polysaccharides, lipids, proteins, and chromophoric dissolved organic matter (Sabbaghzadeh et al., 2017; Cunliffe et al., 2013). It means, they are consisted of dissolved, colloidal and particulate matter. Thus, not to dispose the absorbing and fluorescent matter involved into a gel structure we don't filtrate the samples. In the manuscript we present the results of absorption and fluorescence indices based on CDOM absorption spectra and FDOM 3D fluorescence spectra, collected during three cruises and carried out on the unfiltered samples.

Thus, the measured spectra of unfiltered water are distorted by the effects of scattering and refraction on the large molecules of particle matted. The absorption spectra are the mostly disrupted by scattering in short wavelength-UV, due to small protein-like particles, and in long wavelength - visible range, due to scattering on particle molecules of phytoplankton and the phytoplankton's absorption from at 430 nm to 670 nm). But the applied absorption indices (spectral slope and spectral slope ratio) are calculated in the spectral region that hardly overcome the discussed (mentioned above) spectral ranges. Moreover, the fluorescent spectra may be disordered by Rayleigh and Mie scattering on particle matter. However, in the first step of the 3D fluorescence spectra analysis the scattering effects are subtracted.

The test on differences between the filtered and unfiltered water were performed in September'16. The Figure 1. presents the results of CDOM absorption coefficient at seven wavelengths (290, 300, 310, 330, 355, 375 and 412 nm) for the samples collected in a) the SML and b) SS. The Figure 2. presents the dependence between salinity and FDOM intensity at main FDOM peaks, [R.U.] for a) the SML and b) SS. The Fig. 3 shows a dependence between $a_{CDOM}$(355nm), [$m^{-1}$] and FDOM intensity at peaks [R.U.] for a) unfiltered and b) filtered samples from the SML and SS. The absorption spectra show the differences in the values of the absorption coefficient, between filtered and unfiltered probes, from about 2 $m^{-1}$ to 0.1 $m^{-1}$ for estuary waters to the open sea, respectively, both, for the SML than SS. However, the absorption indices are calculated on the base of the shapes of the spectra (in other words: are based on the relative differences between the values of $a_{CDOM}(\lambda)$), therefore the filtration should not affect their results. Moreover, the filtration changes the fluorescence spectra (Fig. 2) for a component T (protein-like) only. However, the differences are the same for the SML and SS.

Thus, in the future we plan preparing both, filtered and unfiltered samples for laboratory analysis, to compare the results of the absorption and fluorescent indices, calculated for both, filtered and unfiltered water probes. However, based on the tests, we assume that the results of absorption and fluorescent indices for unfiltered samples can be apply as the information about the properties of chromophoric organic matter contained in the gelatinous structures of surface biofilm.

[Figure]

Fig.1. Dependence between salinity and $a_{CDOM}(\lambda)$, [m$^{-1}$] for filtered and unfiltered samples from a) the SML and b) SS.

[Figure]

Fig. 2. Dependence between salinity and FDOM intensity at peaks, [R.U.] for filtered and unfiltered samples from a) the SML and b) SS.

Line 8: "smaller and smaller" is meaningless. Provide numbers and statistics for the decrease a sthe information is hard to extract from the figure

**Response:**

To illustrate the decreasing differences between the values of absorption coefficients for SML and SS with increasing salinity, see Figure 3. Fig. 3 presents the dependence between salinity and the CDOM absorption coefficients, at several wavelengths: 254, 355 and 412 nm for\the SS and SML, upper and lower graphs, respectively. The values of $a_{CDOM}(\lambda)$ decrease with salinity, in both: the SML and SS. However, the values of linear regression coefficients, for this dependence, are higher in the SML than in SS. Thus, for low salinity the values of $a_{CDOM}(\lambda)$ in the SML are higher than in SS, while with increasing salinity the values of $a_{CDOM}(\lambda)$ decrease with increasing salinity and the difference between the results of $a_{CDOM}(\lambda)$ for the SML and SS decrease as well.

[Figure]

Figure 3. Dependences between salinity and CDOM absorption coefficient at several wavelengths.

Line 18: W1 is near-shore, not open sea, correct?

**Response:** Yes, station W1 is near the river outlet, while W9 is in open sea.

Line 24-25: The authors assumes correlation and linear regression is the same. That is incorrect (please refer to textbook for statistics). In statistics, correlation is described as correlation coefficient r, and

not as coefficient of determination (r2 commonly used in regression analysis). Also provide p values to describe the significance of the correlation. Linear regression requires an independent and dependent variable, which is not the case here.

**Response:** I describe the relation between salinity and several absorption indices and use the regression coefficient to show the force of the dependency between the changes of one and the other variable. Therefore I should use "determination coefficient' when I describe the linear relation (and values of the regression coefficient) between the variables. Anyway, the correlation coefficient, and $R^2$, the coefficient of determination, are both measures of how well the regression model describes the data. R values near 1 indicate that the equation is a good description of the relation between the independent and dependent variables. In somehow I'd like to make prediction of the changes of absorption indices but my study are based on in-situ data and my database allow just working out the results and show the relations between the data.

For the calculations of the linear regression of the variables I applied the confidence interval 95%, so p values were smaller than 0.05 – it means that probability of being wrong in concluding that there is an association between the variables. The smaller the P value, the greater the probability that there is an association.

Line 27-38: " the processes go faster in SML than in SS." i don't understand. What processes? Why faster? please explain.

**Response:** If a linear regression coefficient for a dependence described by the variables  has a greater value in SML than in SS, it means that a proportion between the variables is higher in SML than in SS. And this situation means that the changes of a parameter along Axis Y vary faster with the values along Axis X, while these changes have nothing to do with time. Or, in the other words the relationship is stronger for SML and weaker in SS.

Line 14: see above regarding regression vs correlation

**Response**: I write about the relation between the changes of both salinity and fluorescence intensity, emitted by the main component of marine FDOM (A, C, M and T).

Line 16-19: Are the differences significantly differences? From figure 6, it seems some of the comparision of R.U. are not significantly different, but it requires statistical test and p values, which the authos should describe.

**Response:**

Firstly, the ANOVA test was applied for the results presented on Figure 6. The differentiation factor for the results presented on the figure is the level of sampling: SML or SS . The null hypothesis, $H_0$, is that the levels of sampling are meaningless (irrelevant), while the alternative hypothesis, $H_1$, is that the levels of sampling are significant. When we apply the Standard Deviation Statistics, proposed by Fisher, for significance level 95% (2 standard deviations) we obtain that the difference presented as the bars on figs 6 are statistical insignificant. Thus, from the ANOVA test we received the results that we cannot reject the null hypothesis, $H_0$. Thus, the results presented on the figure 6 might be a completely random distribution. However, in spite of the p-values indicate no statistical significance, one can see on the graphs that the values for the SML are always higher than for the SS. Hence, the distinguish between the results for the SML and SS exist. What is more, the differentiation factor is the level of sampling.

Then, we applied the ANOVA test for figure 7, where the differentiation factor for the results is salinity. The null hypothesis , $H_0$, is that the different salinities are irrelevant (insignificant). The alternative hypothesis is that the salinity regimes are significant. The ANOVA test gives information that we can reject the null hypothesis. Thus, the salinity regimes for the results SML and SS are statistically significant.

Figure 6 and 7: Slightly confusing as in Figure 6 authors grouped SML and SS in a single plot, but in Figure 7 grouped between < 7 PSU and > 7 PSU.

**Response:** Figure 6 present the difference between fluorescence intensities for the SML and SS. While, in Figure 7 the results of percentile contribution are presented in different salinity regimes, for the SML and SS, separately. The left and right graphs, for SS and SML, respectively, show the wider range of changes of percentile contribution of the all FDOM components in the SML than SS. The statistical significance was obtain for such a presentation of the results, where the differentiation factor was salinity. Moreover, the results of percentile composition of the main components of marine FDOM (included in Figure 7) can be presented in the same way as in the results in Figure 6 (in the Manuscript), Figure 4.

[Figure]

Figure 4. Dependence of percentage contribution of the main FDOM components in SML and SS as the box plots for (a) coastal waters (salinity <7) and (b) open sea (salinity >7).

Page 12:
Line 16/17: is this statistically different based on a significance level of 95%?

**Response:** The calculations of the linear regression were made by Sigma Plot Toolbox with the confidence interval 95%. The calculations give the values 'a', 'b' and '$r^2$'.

Page 13 - Discussion is short (compared to the Results) and without a single reference to the literature. The authors need to clearly define section Results and Discussion, or combined both if guidelines of the journal allows it. More importantly, the authors need to discuss their results with findings from the literature, e.g. in terms of relevant processes at the sea surface such as light penetration and photochemistry.

**Response:** I put the changes into the manuscript.

Violetta Drozdowska

Institute of Oceanology

Polish Academy of Sciences, u. Powstańców Warszawy 55,

81-712 Sopot, Poland

June 20, 2017

Dr Oliver Zielinski,

Mrs  Natascha Töpfer,

Editors

Ocean Science,

Re: Response to reviewers comments on manuscript by Drozdowska et al., entitled "Title: Study on organic matter fractions in the surface microlayer in the Baltic Sea by spectrophotometric and spectrofluorometric methods."  submitted to Ocean Science and coded OS-2017-4

We thank the reviewers for their constructive comments. We have followed their guidance, and rewritten parts of the manuscript to place the work in better context. Especially the Discussion is now much improved thanks to the suggestions of the Reviewers. We have also gone through the text thoroughly to make any edits to the text to improve the flow and any grammatical errors we found that could be corrected. With this exercise we have also rewritten the key points and abstract to better highlight the main findings of this work.

The detailed comments to the Reviews are given below. After each Reviewers comments, our responses are written using different font face and start with **Response:**

Detailed Response to review by Reviewer #2:

General comments

In the manuscript entitled "Study on organic matter fractions in the surface microlayer in the Baltic Sea by spectrophotometric and spectrofluorometric methods" authors pay attention to an important issue regarding to the influence of surfactants on the physical processes occurring in the sea surface layers. In the manuscript authors present an important results of absorption and fluorescence for samples collected from a surface microlayer (SML) and subsurface layer (SS) in the Batic Sea both in the open sea and near-shore. Based on absorption and fluorescence measurements authors determined several parameters to describe the changes of organic matter and discuss the processes occur in the sea surface layers.

The manuscript consists of 5 sections. In section 1 authors introduce the reader to the issue. In Section 2, authors present collection and characterization of samples, studied area and specification of measurements. Moreover, in this section authors present the detailed description of several specific absorption and fluorescence indices. Next, in section 3 authors present detailed description of obtained results taking into account the specific absorption and fluorescence indices and the relationships between them. Next, authors discuss obtained results in the Section 4 and finally the authors conclude the manuscript in Section 5.

The manuscript reports findings that are interesting for future work in ocean optics. Manuscript has scientific weight.

In my opinion the manuscript require several corrections to be suited for publication. The suggested corrections before the publication of the manuscript were mentioned below.

**Response:** We thank the Reviewer #2 for appreciating our work.

Detailed comments:
1. The name of the Section 2 "Method" should be specified. Now the name of this section suggests about description of used method, however authors describe in this section several issues: the used material, studied area and specification of measurements or specific absorption and fluorescence indices.

**Response:** We agree with the Reviewer #2 remarks. The Section 2. (titled: "Methods") contains the information on many aspects of the marine and laboratory measurements: a study area and sampling methodology, the equipment and laboratory measurements as well as the calculations of the collected dat. Therefore the name of the Section 2. will be changed into **"Measurements"** and, the Section 2. will be divided into 3 subsections titled : **2.1** SML sampling; **2.2** Laboratory spectroscopic measurements of CDOM and FDOM optical properties; **2.3** Optical indices of CDOM and FDOM used for calculations.

2. I suggest, that better would be if authors move the subsection 2.3 "CDOM and FDOM optical properties" and described this in separate section "Optical indices used for calculations" or "Optical indices of CDOM and FDOM used for calculations" with two subsections: absorption indices and fluorescence indices.

**Response:** We agree with the Reviewer #2 remarks. The Section **2.3** "CDOM and FDOM optical properties" is consisted of two parts: the first one deals with the absorption spectra analysis and calculations of the absorption indices and the second one is dedicated to fluorescence data. Hence, the Section **2.3** will be divided into two subsections: **2.3.1** Absorption indices and **2.3.2** Fluorescence indices.

3. I think that better would be, if all data of calculated optical indices have been collected in one table.

**Response:** In our opinion it would be better to leave two separate tables to follow the text with the data included in the tables. The data included in the Table I refer to the results of absorption analysis, discussed in section 2.3.1, while in the Table II to the results of the fluorescence analysis described in the section 2.3.2. Moreover, the results included in the Table I are calculated for the first and last station only, along the sampling transect during three research cruises. While, the median of the fluorescence indices presented in Table II are calculated for the group of samples collected in water of less salinity ($< 7$) and of higher salinity ($> 7$).

4. The English language of the manuscript is good. However, several sentences are unclear or contain colloquial phrases, for example: page 7 line 8 "...become smaller an smaller.." better would be "...decreasing..." page 12 line 2 "...the biggest relative changes..." better would be "...the highest relative changes.." page 13 line 3 "...shorter wavelength..." better would be "...lower wavelength.." page 15 line 7 "What is more..." better would be "Moreover.." I think that a little English correction can improve quality of the manuscript.

**Responses:** The sentences has been corrected and rewritten according to Reviewer #1 request: page 7 line 8, there was: Moreover, with an increase of a distance from the river outlet, the intensity of light absorption is decreasing significantly and the differences between the SML and SS become **smaller and smaller**.

There is: Moreover, with an increase of a distance from the river outlet, the intensity of light absorption is decreasing significantly and the differences between the SML and SS **decreasing as well**.

 there was: Hereby, the **biggest** relative changes of the FDOM component composition, along the transect from the Vistula River outlet to Gdansk Deep, were recorded for component T, both in SML and SS (about 18.5 % and ~12.3 %, respectively), while the relative changes of  A, C and M components were: 4.1, 8.1 and 2.6 % in SML and 1.9, 3.1 and 4.7 % in SS, respectively.

It is: Hereby, the **highest** relative changes of the FDOM component composition, along the transect from the Vistula River outlet to Gdansk Deep, were recorded for component T, both in SML and SS (about 18.5 % and ~12.3 %, respectively), while the relative changes of  A, C and M components were: 4.1, 8.1 and 2.6 % in SML and 1.9, 3.1 and 4.7 % in SS, respectively.

Response:  In my opinion, 'lower' and 'shorter' have the same meaning with respect to the limit of wavelength range of light and may be used interchangeable. However, we agree with the Reviewer, that in this context: 'low' fits better.

There was: However the photochemical degradation processes, resulting in a decrease in the mass of molecules and an increase of concentration of low molecular-weighted molecules, are much more spectacular in a **shorter** wavelength range and are held primarily in the surface microlayer, SML.

There is: However the photochemical degradation processes, resulting in a decrease in the mass of molecules and an increase of concentration of low molecular-weighted molecules, are much more spectacular in a **lower** wavelength range and are held primarily in the surface microlayer, SML.

 there was: **What is more**, a decreasing of DOM concentration with salinity occurs faster in SML than in SS.

There is: **Moreover**, a decreasing of DOM concentration with salinity occurs faster in SML than in SS.

5. The data presented in tables are unnecessarily duplicated in the text, for example: - page 7 line 15-17 - duplicated data from Table 1 or - page 10 line 17-19 duplicated data from Table 2, instead the duplication, the authors should discuss this data.

**Response:** We agree with the Reviewer #2 that we should add some comments in this paragraph. Thus, some comments was added. However we decided to leave the text, with the numbers, unchanged.

The additional sentences: The higher values of $S_R$ in the SML in the open sea waters, mean the smaller size of CDOM that may exist due to a photodegradation process (Helmes et al., 2008). While the lower values of $S_R$ in the SML in the vicinity of the river outlet may mean the forming of the surface structures from the hydrophobic molecules coming with freshwater (Cuncliffe et al., 2009).

The legend in the Figure 3 was corrected. There was:

[Figure]

There is:

[Figure]

**7. Fig. 7 - no description of X-axis**

We corrected the Fig 7 by adding the description : "FDOM components", to the Axis X.

There was:

[Figure]

It is:

[Figure]

**8. Page 7 line 19 - W1 station describes the area near Vistula River outlet not open sea**

**Response:** The sentence has been corrected. There was:  However in **W1** (the open sea) the differences were 3.1, 1.5 and 3.5 %, while in **W9**: 10.5, 5.4 and 11.9 %.

There is: However in **W9** (the open sea) the differences were 3.1, 1.5 and 3.5 %, while in **W1**: 10.5, 5.4 and 11.9 %.

**Response:** The sentence has been corrected. There was: A spectral slope coefficient ratio, $S_R$, ($S_{274-295}$ to $S_{350-400}$) is negatively correlated with molecular weight of CDOM in humic substances.

There is: A spectral slope coefficient ratio, $S_R$, ($S_{275-295}$ to $S_{350-400}$) is negatively correlated with molecular weight of CDOM in humic substances.

The references list has been also updated.

Violetta Drozdowska

Institute of Oceanology

Polish Academy of Sciences, u. Powstańców Warszawy 55,

81-712 Sopot, Poland

June 20, 2017

Dr Oliver Zielinski,

Mrs  Natascha Töpfer,

Editors

Ocean Science,

Re: Response to reviewers comments on manuscript by Drozdowska et al., entitled "Title: Study on organic matter fractions in the surface microlayer in the Baltic Sea by spectrophotometric and spectrofluorometric methods." submitted to Ocean Science and coded OS-2017-4

We thank the reviewers for their constructive comments. We have followed their guidance, and rewritten parts of the manuscript to place the work in better context. Especially the Discussion is now much improved thanks to the suggestions of the Reviewers. We have also gone through the text thoroughly to make any edits to the text to improve the flow and any grammatical errors we found that could be corrected. With this exercise we have also rewritten the key points and abstract to better highlight the main findings of this work.

The detailed comments to the Reviews are given below. After each Reviewers comments, our responses are written using different font face and start with **Response:**

Detailed Response to review by Reviewer #3:

General comments:

The manuscript titled "Study on organic matter fractions in the surface microlayer in the Baltic Sea by spectrophotometric and spectrofluorometric methods" by Drozdowska et al add to our knowledge about optical parameters of the microlayer and surface layer. In my opinion, it is a step towards remote sensing of microlayer properties, something that would be extremely helpful for studying its effect on air-sea interaction fluxes. I believe the manuscript documents well what and how has beed measured. I recommend publishing it after minor revision.

**Response:** We thank the Reviewer #3 for appreciating our work.

The open review process of the EGU journals has both advantages and disadvantages.
However, the fact that I see the previous two reviews makes it easier for me because I do not need to repeat what has already been told. So agreeing with most of the commands of my respectable anonymous peer-review colleagues, I will just comments on thing which I did not see in their comments.

**Response:** The authors agree with most of the comments included in the reviews #1 and #2 and #3 and the prepared final manuscript has been corrected according to the advice given by Reviewers #1 and #2 and #3.

Detailed remarks:

The fluorescence intensities A, C M and T should be explained in the abstract. Something simple like "fluorescence intensities at Coble classification peaks" should be enough to give some hint to the reader what they are.

**Responses:** The main fluorophores included in marine CDOM molecules were, the first time, classified by Coble (1996). The maxima of the main fluorescence peaks of marine CDOM are different for freshwater, open sea, coastal and estuary waters and are the subject of many studies. Thel information about naming of A, C, M and T as "fluorescence intensities at Coble classification peaks" is put to the abstract.
It was: . Several absorption ($E_2$:$E_3$, S, $S_R$) and fluorescence (fluorescence intensities at peaks: A, C, M, T, the ratio (M+T)/(A+C), HIX) indices of colored and fluorescent organic matter (CDOM and FDOM) helped to describe the changes in molecular size and weight as well as in composition of organic matter.

It is: . Several absorption ($E_2$:$E_3$, S, $S_R$) and fluorescence (fluorescence intensities at **Coble classification** peaks: A, C, M, T, the ratio (M+T)/(A+C), HIX) indices of colored and fluorescent organic matter (CDOM and FDOM) helped to describe the changes in molecular size and weight as well as in composition of organic matter.

Units in the figures should be presented in [ ] braces.

**Response:** The all units in the all figures were put into the brackets [...] were, except Fig. 1. The figures: 2,3,4,5,6 and 7 are:

[Figure]

[Figure]

[Figure]

[Figure]

[Figure]

Date format in Table 1 is certainly not something most English native speakers will recognize. Because of the US/UK dichotomy (09/11/2001 versus 11/09/2001), I suggest using month names explicitly (11 September 2001). The hyphen in "October'2015" is not necessary (at least in two places). One uses it only to shorten the year (October '15).

**Response:** The name of date in Table 1 and in the text were written explicitly.

There was:

| | A slope ratio – $S_R$ (= $S_{275-295}/S_{350-400}$) | | | | | |
|---|---|---|---|---|---|---|
| | $S_R$ - for SS | | | $S_R$ - for SML | | |
| | 28th IV 2015 | 15-16th X 2015 | 11th IX 2016 | 28th IV 2015 | 15-16th X 2015 | 11th IX 2016 |
| W1 | 1.58 | 1.16 | 1.61 | 1.43 | 1.10 | 1.35 |
| W9 | 1.30 | 1.33 | 1.40 | 1.34 | 1.35 | 1.45 |

There is:

| | A slope ratio – $S_R$ (= $S_{275-295}/S_{350-400}$) | | | | | |
|---|---|---|---|---|---|---|
| | $S_R$ - for SS | | | $S_R$ - for SML | | |
| | 28 April 2015 | 15-16 October 2015 | 11 September 2016 | 28 April 2015 | 15-16 October 2015 | 11 September 2016 |
| W1 | 1.58 | 1.16 | 1.61 | 1.43 | 1.10 | 1.35 |
| W9 | 1.30 | 1.33 | 1.40 | 1.34 | 1.35 | 1.45 |

I commend the authors for using unitless practical salinity (as all the relevant standards have it). However, the word "practical" should be added somewhere before salinity to make it obvious that the salinity was not absolute.

**Response:** The authors agree with the Reviewer #3 remark and just put the name "Practical salinity" – as a description of the X axes – into the figures: Fig.3 and Fig.5.

There is: Figure 3

[Figure]

There is Figure 5:

[revised manuscript text omitted]

Pempkowiak, 2015).

Surfactants comprise a complex mixture of different organic molecules of amphiphilic and aromatic structures (with hydrophobic and/or hydrophilic heads) rich in carbohydrates, polysaccharides, protein-like and humus (fulvic and humic) substances (Williams et al., 1986; Ćosović and Vojvodić, 19968; Cuncliffe et al, 2011)Surfactants comprises a mixture of organic molecules rich in lipids (fatty acids, sterols), polymeric and humus which proportions determine the various properties of the SML. Some dissolved organic compounds possess, especially fulvic and humic substances, the optically active parts of the molecules that absorb the light, called i.e. chromophores, that absorb the light energy (CDOM, *chromophoric* dissolved organic matter), and fluorophores, that absorb and emit the light (FDOM – fluorescent dissolved organic matter). Due to the complexity and variability of the compositional variability of the dissolved marine organic matter mixture, the the absorption and fluorescence (excitation-emission matrix) spectroscopy (Stedmon et at, 2003; Hudson et al., 2007; Coble, 2007). best were found as fast and reliable available tool methods (fast and reliable) to for detection and identifyication the of the dissolved
* * *
**Sformatowano:** Nie Wyróżnienie

**Z komentarzem [A3]:** ???? – czy nie miało być aliphatic ?- Jest ok. „Amfifilowe", to takie hydrofobowo-hydrofilowe.

**Z komentarzem [A4]:** Niedobre słowo – czy chodziło o podstawnik – jeśli tak to trzeb znaleźć odpowiednie słowo w słowniku. – W literaturze przedmiotu również używa się tego określenia „head"

organic matter in seawater (Stedmon et at, 2003; Hudson et al., 2007; Coble, 2007; Jørgensen et al., 2011). . is the absorption and fluorescence (excitation emission matrix) spectroscopy (Stedmon et at, 2003; Hudson et al., 2007; Coble, 2007). A unique structure of the energy levels of these organic molecules results in a specific spectral distribution of the light intensities absorbed and emitted by the molecules. Hence, the aAbsorption and fluorescence spectra of specific organic compounds groups may allow the identification of the sources transformations of dissolved organic matter (Coble, 1996; Lakowicz, 2006). Several indices describing the changes of a concentration (citationBlough and Del Vecchio, 2002), a molecular weight (Peuravuori and Pihlaja, 1997)citation), a composition of CDOM/FDOM (Stedmon and Bro, 2008; Boehme and Wells, 2006citation) and a rate of degradation processes (Milori et al., 2002; Glatzel et al., 2003; Zsolnay, 2003citation) can be calculated from The analysis of the CDOM absorption and 3D FDOM fluorescence excitation and emission matrix fluorescence spectra EEMs, that could be useful to study dissolved organic matter dynamics and composition in surface micro layer. enabled to calculate several indices describing the changes of a concentration, a molecular weight, a composition of CDOM/FDOM and a rate of degradation processes of the organic matter occurring in the study surface layers.

There are many applications Recent advances in applications of the absorption -and fluorescence spectroscopy in environmental studies on oceanographic aquatic dissolved organic matter both in fresh and marine environments and engineered water systems have been summarized in numerous text books and review papers (e.g. Coble, 2007; Hudson et al., 2007; Ishii and Boyer, 2012; Andrade-Eiroa et al., 2013ab; Nelson and Siegel, 2013; Coble et al., 2014; Stedmon and Nelson, 2015). The humic substances contribute significantly both to CDOM pool in the water column as well as to surfactants concentrations especially in coastal ocean, estuaries and semi-enclosed marine basin that are impacted by terrestrial runoff and marine traffic. Therefore optical methods could be usedd efficiently for determination of natural and anthropogenic organic surface active substances in SML (Drozdowska et al. 2013; Drozdowska et al., 2015;, Pereira et al., 2016; Frew et al.,2004; Zhang et al., 2009; McKnight et al., 1997; Guéguen et al., 2007. Baszanowska [22]) studies on mixing water masses locally, e.g. in estuaries (Williams et al., 2010) and in global scale (Jorgensen et al., 2011). The studies were conducted in various natural waters as e.g. Chinese lakes (Zhang et al 2013; Chen et al., 2011), Indian Ocean (Chari et al., 2012), American

Sformatowano: Angielski (Stany Zjednoczone)

Sformatowano: Angielski (Stany Zjednoczone)

Z komentarzem [A5]: Jakieś inne międzynarodowe publikacje???

estuaries (Glatzel et al., 2003; McKnight et al., 1997; Moran et al., 2000) and in studies on dilution sea basi ns,

Baltic Sea is a semi-enclosed marine basin with annual riverine discharge reaching ca. 0.XX5 $10^3$x km$^3$ of fresh water (Leppäranta and Myrberg, 2009),. Maximum freshwater runoff occurs in April/May. The fresh water carries both high concentrations of CDOM (Drozdowska and Kowalczuk, 1999; Kowalczuk, 1999; Kowalczuk et al., 2010;, Ylostallo et al., 2016) and substantial loads anthropogenic pollutants and inorganic nutrients (Drozdowska et al., 2002; Pastuszak et al., 2012) that stimulates phytoplankton blooms, This marine basin is also impacted by significant pollution caused by the high marine traffic (Kkonik and Bradtke, 2016 XX). such as the Baltic (Kowalczuk et al., 2010; Drozdowska et al., 2002) and Arctic (Gueguen et al., 2007) that considered the differences in FDOM components from the rivers, lakes and inland water.

This The main goal of this study was i) to distribution of concentration of specific CDOM/FDOM components in the SML and subsurface waters (SS - 1 m depth) in the salinity gradient along a transect from the Vistula River mouth to Gdansk Deep, Gulf of Gdansk, Baltic Sea; ii) observe the compositional changes of CDOM/FDOM derived from changes of spectral indices calculated from absorption and EEM spectra; paper is focus on iii) describe and iii) distinguishing fate and concentration of specific CDOM/FDOM components of organic matter to detect and describe the processes that lead to observed differences in CDOM/FDOM concentration and that composition in the SML and SS along sampled transect. occur in the sea surface microlayers (SML) and in subsurface layers (SS), a depth of 1 m. Research are based on the absorption and fluorescence spectra and several absorption and fluorescence indices. Investigations concern the region of Gulf of Gdansk, along a transect from the Vistula River outlet (the biggest Polish river) to open sea.

**2 Measurements Methods MeasurementsMaterials and methods**

**2.1 Materials and study areaSML sampling**

Sample collection for spectroscopic characterization Research to identifyof the dissolved organic matter contained in the SML and SS, that could be regarded as proxy for marine surfactants were conducted during three research cruises of r-/-v y 'Oceania' in April and October (two cruises in 2015 and one in September 2016). The study was conducted Measurement of physical parameters of sea water and samples collection were performed at nine stations along the transect 'W' - from the mouth of the Vistula River, W1, along the

Gulf of Gdansk to the Gdansk Deep in the open sea, W9, (Figure 1). The study areaGulf of
Gdansk is under direct influence of the main Polish river system, Vistula, which drains the
majority of Poland (Uścinowicz, 2011). Meteorological observations (wind speed and wind
direction, and a surface waves high were recorded) and CTD cast with use of the SeaBird
SBE 19 probe was performed The following tasks were performed at every station. Water
samples were collected at SML and SS.: (i) measurement of the hydrophysical parameters
(CTD), (ii) sampling the seawater from SML and SS, (iii) preparation the samples to the
appropriate laboratory tests (filtration and proper maintenance) and (iv) meteorological
observations. The SML Ssampling was carried out when the sea state was 10-4 B only, and
there were no detectable oil spills. The samples were collected from the board of the vessel
(r/y Oceania), that is about 2 m above the sea surface. The sampling was maintained about
15 minutes after anchoring, to avoid the turbulences in the surface layer caused by the screw
and ship movements. We used the Garrett Net, mesh 18, (18 wires per inch), to collect the
samples from the sea surface microlayer, according to the procedure described by Garrett
(1965). The mesh screen is 650 cm x 650 cm, made of metal, and the size of holes is 1 mm,
while the thicknessdiameter of the wire is 0.4 mm. Thus, the thickness of a collected
microlayer is about 0.5 mm. and the efficiency is 60%. On average, 22 such samplings were
required to obtain 1 dm$^3$ of microlayer water. The following sampling procedure was
established. First, the screen was immersed. at an angle of 45^. Then, once totally immersed,
the screen was left under the water until the microlayer had stabilized. Finally, it was
carefully raised to the surface in a horizontal position at a speed of ca 5–6 cm s$^{-1}$ (Carlson
1982). Water was poured from the screen into a polyethylene bottle using a special slit in
the screen frame. The SML samples were collected by the metal Garret's net of 500 μm mesh.
This technique allows collecting water from the top-layer of an approximately 1 millimeter
(Garrett, 1965). In the same places the SS samples from a depth of 1 m were taken by a
Niskin bottle. The unfiltered samples were placed into dark bottles and stored at 4°C.
Collected, unfiltered water samples were stored in amber glass bottles in the dark at 4°C
until analysis in the land based laboratory.During sampling the measurements of temperature
and salinity of a surface layer were conducted. Sampling was carried out when the sea state
was 1-4 B only, and there were no detectable oil spills. Additionally, the meteorological
observations (e.g. recorded wind speed and wind direction and a high of a wave) during
sampling, proved to be valuable in the interpretation of extraordinary results. During
sampling, in two research cruises, at April in 2015 and September in 2016, the wind speed
was almost equally to zero. However, in October in 2015, a northern-west wind was recorded

**Z komentarzem [A6]:** Recenzent chce dokładnego opisu poboru próbek w SML – to jest za mało.

(3-4 B). In October the cruise started after a week-long storm of northerly winds resulting in
the influx of water from the open sea and strong mixing of fresh with coastal and sea water.
That allows the explanation of the surprisingly low concentrations (typical for a salinity
above 7) of organic matter recorded along entirely transect W, even at the vicinity of the
mouth of the Vistula River.

[Figure]

Figure 1 . Measurementsing stations realized sampled during research cruises of
r/vv Oceania: 28th April and 15-16th October in 2015 and 11th September
in 2016.

**2.2.** Laboratory spectroscopic measurements of CDOM and FDOM optical properties
laboratory measurements.

The studies conducted in laboratory are: (i) measurements of absorption and (ii) 3D
fluorescence (EEM) spectra of the surface (SML) and subsurface (SS) samples, from 27
stations. Spectrophotometric and spectrofluorometric measurements of the collected
samples were carried conducted in laboratory the Institute of Oceanology Polish Academy
of Sciences, Sopot, Poland, within a week24 h after the cruise end. Before any spectroscopic
measurements water samples were left to warm up to room temperature. out in 24 hours after
collection without any previous treatment at room temperature.

The main task in our work was to study the luminescent properties of the molecules
that form a surface microfilm. However, As it is well known, tTthe seasurface microlayer is
a gelatinous film created by polysaccharides, lipids, proteins, and chromophoric dissolved organic matter (Sabbaghzadeh et al., 2017; Cunliffe et al., 2013) and . It means, consisted of dissolved, colloidal and particulate matter. Thus, not to dispose the absorbing and fluorescent matter involved into a gel structure we do no't filtrate the samples. In the manuscript the results of absorption and fluorescence indices based on CDOM absorption spectra and

FDOM 3D fluorescence spectra, collected during three cruises and carried out on the unfiltered samples are presented. There were performed tThere were performed the tests on filtrated and unfiltrerated probes, sampled during one cruise (not published). Changes in the absorption spectra resulting from the unfiltering of the samples occur mainly in the short UV

and far VIS range. However, these differences do not cause significant changes in the absorption indices, because they are calculated on the basis of the shapes of the spectra (in other words: are based on the relative differences between the values of $a_{CDOM}(\lambda)$) in the range between the affected ends of the measuring range We obtain that in spite of the differences in the values of the absolute values of the absorption coefficient, between filtered and unfiltered probes, the absorption indices are calculated on the base of the shapes of the spectra (in other words: are based on the relative differences between the values of

$a_{CDOM}(\lambda)$). Moreover, in the studied fluorescence spectra, due to lack of filtration, we obtain a strong elastic and non-elastic scatter band, which, however, is removed in the first step of the analysis., therefore the filtration do not effect their results. Moreover, The the filtration procedure affects changes the fluorescence spectral band (Fig. 2) for a component T

(protein-like) only, that is much effectively retained on the filter, . Hhowever, the differences are the same for the SML and SS. Iit is well known that, filtration separates particulate fraction from dissolved and colloidal ones. On the other hand, during filtration the strongly surface active structures of organic molecules or macromolecules might be retained on the filter by sorption processes (Ćosović and Vojvodić, 1998). Knowing the limitations of the applied procedures, we decide to conduct research on unfiltered water Therefore, the all studied samples are analyzed without filtration(Ćosović and Vojvodić, 1998; Drozdowska et al., 2015). Samples for absorption and fluorescence measurements were treated in the same manner.

CDOM absorption measurements were done with use of Perkin Elmer Lambda 650

spectrophotometers in the spectral range 240 – 700. All spectroscopic measurements were done with use of 10-cm quartz cell and ultrapure water MilliQ water was used as the reference for all measurements. Raw recorded absorbance A($\lambda$) spectra were processed and the CDOM absorption coefficients $a_{CDOM}(\lambda)$ in [m$^{-1}$] were calculated by:

**Sformatowano:** Nie Wyróżnienie

**Sformatowano:** Nie Wyróżnienie

**Sformatowano:** Nie Wyróżnienie

**Sformatowano:** Nie Wyróżnienie

**Sformatowano:** Nie Wyróżnienie

**Z komentarzem [A7]:** Jak uwzgleniłaś rozpraszanie i absorpcję cząstek w pomiarach spektrofotometrem. Jak uwzględniłaś rozpraszanie w pomiarach fluorescencji?

$$aCDOM(\lambda) = 2.303 \cdot A(\lambda)/l \hspace{3cm} (1)$$

[revised manuscript text omitted]